# Tailoring coordination environments of single-atom electrocatalysts for hydrogen evolution by topological heteroatom transfer

Sheng Qian[1], Feng Xu[1], Yu Fan[1], Ningyan Cheng[2], Huaiguo Xue[1], Ye Yuan [3], Romain Gautier [4] ✉, Tengfei Jiang [1] ✉ & Jingqi Tian [1] ✉

The rational design of carbon-supported transition-metal single-atom catalysts requires the precise arrangement of heteroatoms within the single-atom catalysts. However, achieving this design is challenging due to the collapse of the structure during the pyrolysis. Here, we introduce a topological heteroatom-transfer strategy to prevent the collapse and accurately control the P coordination in carbon-supported single-atom catalysts. As an illustration, we have prepared self-assembled helical fibers with encapsulated cavities. Within these cavities, adjustable functional groups can chelate metal ions ($N_x \cdots M^{n+} \cdots O_y$), facilitating the preservation of the structure during the pyrolysis based phosphidation. This process allows for the transfer of heteroatoms from the assembly into single-atom catalysts, resulting in the precise coordination tailoring. Notably, the $Co-P_2N_2-C$ catalyst exhibits electrocatalytic performance as a non-noble metal single-atom catalyst for alkaline hydrogen evolution, attaining a current density of $100 \, mA \, cm^{-2}$ with an overpotential of only 131 mV.

Carbon-supported transition-metal single-atom catalysts ($M-N_x-C$ SACs) exhibit remarkable catalytic activity[1,2]. The presence of functionalized carbon materials with heteroatom (e.g., N, S, P) anchoring sites creating specific microenvironments for metal single atoms plays a crucial role in the performances of SACs[3]. In the bottom-up pyrolysis-based synthesis of SACs, the origin, location, number, and configuration of coordination heteroatoms on carbon determine the structure, chemical environment, and overall catalytic performance of SACs[4]. Recently, introducing P into well-developed $M-N_4$ SACs has been demonstrated to be a very efficient approach to modulate the electron densities of metal centers and enhance catalytic activities[5]. Thus, successful motifs such as $Co(Fe)-N_3P$[6–9] and $Fe-P_4$[10] have been prepared. However, current P-incorporation methods involve co-pyrolyzing P-containing molecules with metallic and carbonaceous precursors[6–12]. Unfortunately, the carbonaceous precursor undergoes severe structure collapse due to its low thermal stability, making it challenging to accurately arrange the heteroatom P in SACs. Consequently, achieving the precise coordination of heteroatoms in SACs remains challenging up to now, and for this reason, the catalytic properties cannot be further optimized.

To resolve this issue, the carbonaceous precursors with uniformly controlled anchoring sites, and more importantly, solid spacing

[1]School of Chemistry and Chemical Engineering, Yangzhou University, 180 Si-Wang-Ting Road, Yangzhou 225002, P. R. China. [2]Information Materials and Intelligent Sensing Laboratory of Anhui Province, Key Laboratory of Structure and Functional Regulation of Hybrid Materials of Ministry of Education, Institutes of Physical Science and Information Technology, Anhui University, Hefei 230601, P. R. China. [3]Key Laboratory of Polyoxometalate and Reticular Material Chemistry of Ministry of Education, Faculty of Chemistry, Northeast Normal University, Changchun 130024, P. R. China. [4]Nantes Université, CNRS, Institut des Matériaux de Nantes Jean Rouxel, IMN, F-44000 Nantes, France. ✉e-mail: romain.gautier@cnrs-imn.fr; jiangtengfei@yzu.edu.cn; tianjq@yzu.edu.cn

structure for metal atoms could be appropriate candidates, combining the advantages of both small organic molecules and large carbon materials such as graphene and carbon nanotube[13,14]. In the past decade, graphene quantum dots (GQDs), which are unique 0D carbon allotropes characterized by nanoscale polyaromatic molecules with $sp^2$ domains on the basal plane and functional groups at the edges, have drawn a lot of attention[15]. A crucial advantage of GQDs as carbon precursors for SAC synthesis include, among others, the presence of graphitic domains to prevent structure collapse during pyrolysis[16]. Furthermore, edge-dispersed functional groups play a significant role by providing anchoring sites for homogeneous loading of metallic species and provide an opportunity to control the chelation environment.

In this context, we have developed a strategy that involves the noncompetitive chelation between heteroatom-containing functional groups (e.g., -OH and -NH$_2$) on GQDs and metal ions, forming π-conjugation-encapsulated cavity-confined metal complexes ($N_x\cdots M^{n+}\cdots O_y$) within GQD assemblies. This self-assembly of GQDs is followed by the preferential substitution of O by P during phosphidation, leading to the formation of SACs. We illustrate this approach with the synthesis of a series of P-coordinated SACs embedded in carbon. By self-assembling GQDs with amino and hydroxyl groups, we achieve diverse carbon superstructures, such as helical fibers and sheets. Meanwhile, the presence of rigid graphitic domains enables the retention of atomic moieties to a significant extent during phosphidation, while the adjustable chelation environment allows for the customized transfer of heteroatoms, such as P and N, into SACs. This enables the systematic tailoring of the coordination environment in Co SACs (Co−P$_x$N$_{4-x}$−C, where x ranges from 0 to 3) with Co−P$_2$N$_2$−C SAC exhibiting superior activity for the electrochemical hydrogen evolution reaction (HER). Furthermore, we predict the generalizability of this strategy not only to a wide range of

metal centers, including transition metals (e.g., Fe, Ni, Cu) and noble metals (e.g., Pt, Pd), but also to coordinated non-metal heteroatoms (e.g., O, S, Se).

## Results

### Design of self-assembled building block-functionalized GQDs

Graphene quantum dots (GQDs) are synthesized via pyrene-based molecular fusion route under alkaline hydrothermal treatment with urea for the amino functionalization[17]. Detailed characterizations are shown in Supplementary Figs. 1 and 2. The spectral analysis demonstrate the peripheral functionalization by both amino and hydroxyl groups with a ratio of *ca.* 1:2 (A/H = 0.54). This ratio could be readily adjusted ranging from 0 to 0.69 by varying synthetic parameters (see Methods for details, Supplementary Fig. 3, Table 1).

The self-assembly of such amino and hydroxyl functionalized GQDs (NGQDs) in alkaline media leads to the formation of helical fibers. Microscopic characterization using scanning electron microscopy (SEM) and transmission electron microscopy (TEM) reveals that these fibers exhibit lengths on the order of tens of microns and diameters ranging from 60 to 100 nm. Notably, the helical pitch is consistently around 80 nm (Fig. 1a–c). HRTEM analysis demonstrates that such helical fibers consist of crystal domains with interplanar spacing of 0.21 nm corresponding to the graphene (100) planes, and characteristic (002) layer spacing of 0.34 nm which matches that of graphitic carbon (Fig. 1d)[17]. The characteristic (002) reflection peak exhibits a positive shift in the helical fibers compared to that of NGQDs (Supplementary Fig. 4), indicating intermolecular π-stacking induced contact packing of $sp^2$ planes during the assembly process[18]. Atomic force microscopy (AFM) imaging reveals the fibrous nature of the structures, with an average height of 9.0 nm (Fig. 1f), which is approximately three times the lateral size of a single NGQD molecule. This suggests that the fiber is composed of at least three twisted sub-fibers that pack perpendicularly

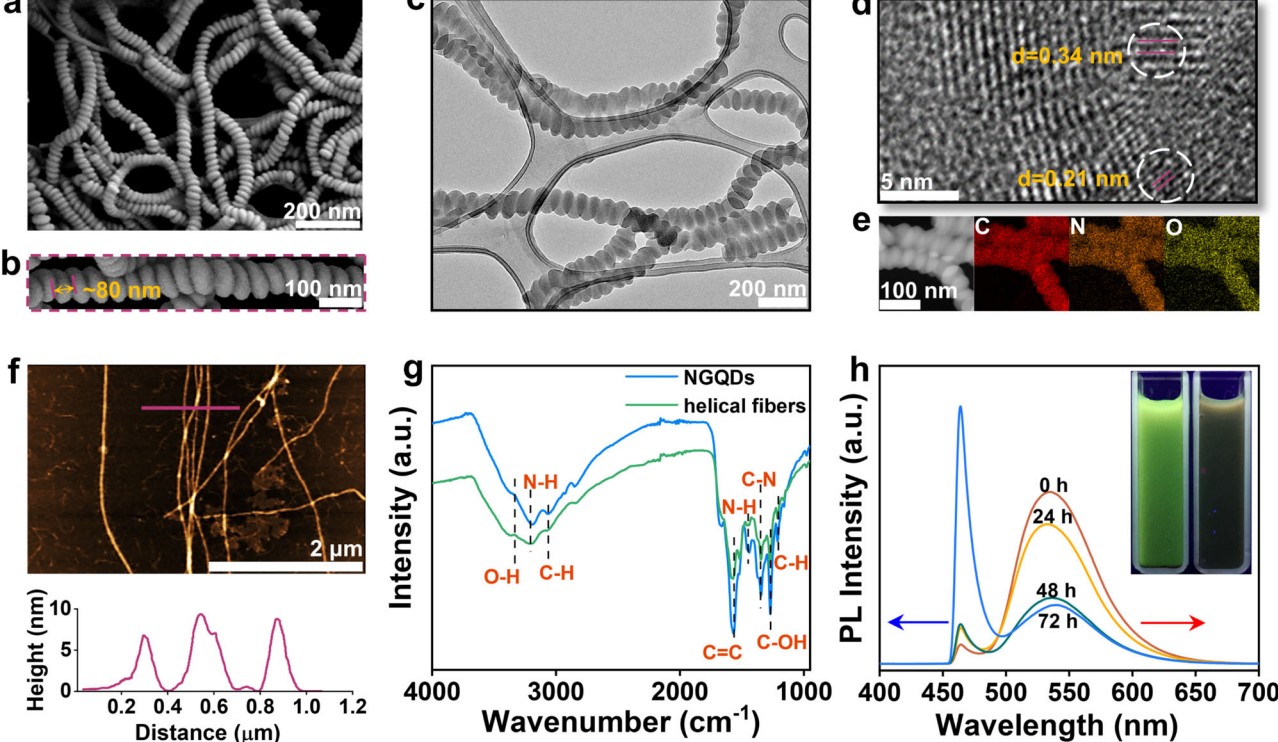

**Fig. 1 | Structure characterizations of the helical fibers.** Low-magnification (**a**) and high-magnification (**b**) SEM images of the helical fibers. TEM image (**c**) and high-resolution TEM image (**d**) and STEM and elemental mapping images (**e**) of the helical fibers. **f** AFM image coupled with height-profile of the helical fibers. **g** FT-IR

spectra of NGQDs and helical fibers. **h** Time-dependent PL spectra during the self-assembly of NGQDs, inset shows the optical photographs taken under UV (365 nm) and in dark.

to the substrate. Furthermore, X-ray photoelectron spectroscopy (XPS) analysis demonstrates similar results for NGQDs and helical fibers (Supplementary Fig. 5). These results indicate that the NGQDs and helical fibers are structurally related.

Based on the above characterizations, the formation of helical fibers in alkaline media can be attributed to the face-to-face alignment of NGQDs facilitated by dominant π–π interactions, which is further evidenced by Raman and FT-IR analysis. The calculated $G$ (1596 cm$^{-1}$)/$D$ (1354 cm$^{-1}$) ratio of helical fibers (1.04) is higher than that of NGQD molecules (0.97), indicating a more pronounced extent of π-π stacking (Supplementary Fig. 6)[19]. In the FT-IR spectra (Fig. 1g), the absorption peak at 1569 cm$^{-1}$ identified as C=C vibration in the $sp^2$ domain of NGQDs blue shifts with a sharp decrease in intensity in the assembly, providing compelling evidence of intensive π-stacking interactions within the helical fibers[20].

Time-dependent photoluminescence (PL) spectra of the NGQDs dispersion were recorded (Fig. 1h, Supplementary Fig. 7), with the observation of two bands at $\lambda_{max}$ = 465 nm (band A) and $\lambda_{max}$ = 535 nm (band B)[21,22]. As explained by the molecular model, the NGQD has a size of 2.3 × 2.6 nm$^2$, with –NH$_2$ and –OH groups located on the edges (Supplementary Fig. 8). During the self-assembly, the central $sp^2$ domains come into contact with those of neighboring NGQDs through π–π stacking interactions resulting in the red shift observed in band B. In contrast, the peripheral $sp^2$ domains interact weakly through hydrogen bonding, leading to an insignificant shift in band A[23]. This observation confirms the dominant role of π-π stacking in the formation of helical fibers. The bathochromic shift of the emissions is consistent with the exciton coupling of the aromatic units due to the formation of H-aggregates[24] (inset in Fig. 1h). As we described the observation of the red shift in band B during self-assembly, we mainly focused on the comparison between the final and initial states, (i.e. between the 72 h sample and the 0 h sample). One can also observe a

4–7 nm blue shift in band B for the 24 h sample (Fig. 1h and Supplementary Fig. 7a), which is attributed to the increased twisted deformation (Corresponding SEM images are shown in Supplementary Fig. 7b). Such a twisted deformation is believed to be due to a slight change in the molecular packing with a slight separation of central $sp^2$ domains upon molecular sliding from the most stable $J$-aggregated state[25,26].

It is noteworthy that only NGQDs with optimized A/H ratios resulted in the formation of helical fibers. High A/H ratios result in non-regular assemblies while low A/H ratios result in bulk aggregates (Supplementary Fig. 9). Too diluted or too concentrated NGQDs solutions failed to form helical-fiber-like assemblies (Supplementary Fig. 10), indicating that the helical fibers are thermodynamically stable products.

## pH-dependent self-assembly of NGQDs

The self-assembly of aromatic molecules is known to be influenced by pH, as it affects the strength of non-covalent interactions such as π-π stacking, hydrogen bonding, and electrostatic interactions[27]. Assembling NGQDs in acidic solutions, e.g., pH=3.5, leads to the formation of nanosheets with lateral sizes in the range of several tens of micrometers (Supplementary Fig. 11a, b). Detailed characterizations confirmed that the nanosheets share similar structural components with the helical fibers but different molecular alignment (Supplementary Fig. 11c–f, Fig. 12).

A scheme is proposed for the self-assembly of NGQDs at different pH (Fig. 2a). The formation of assemblies originates from 1D or 2D nuclei with distinct arrangements of NGQD molecules at high or low pH to further grow into helical fibers or nanosheets, respectively[28]. Molecular dynamics (MD) simulations reveal that, under pH 9.5, the initial NGQDs preferentially construct a columnar nuclei where adjacent NGQD planes were positioned at a distance of 3.6 Å through face-

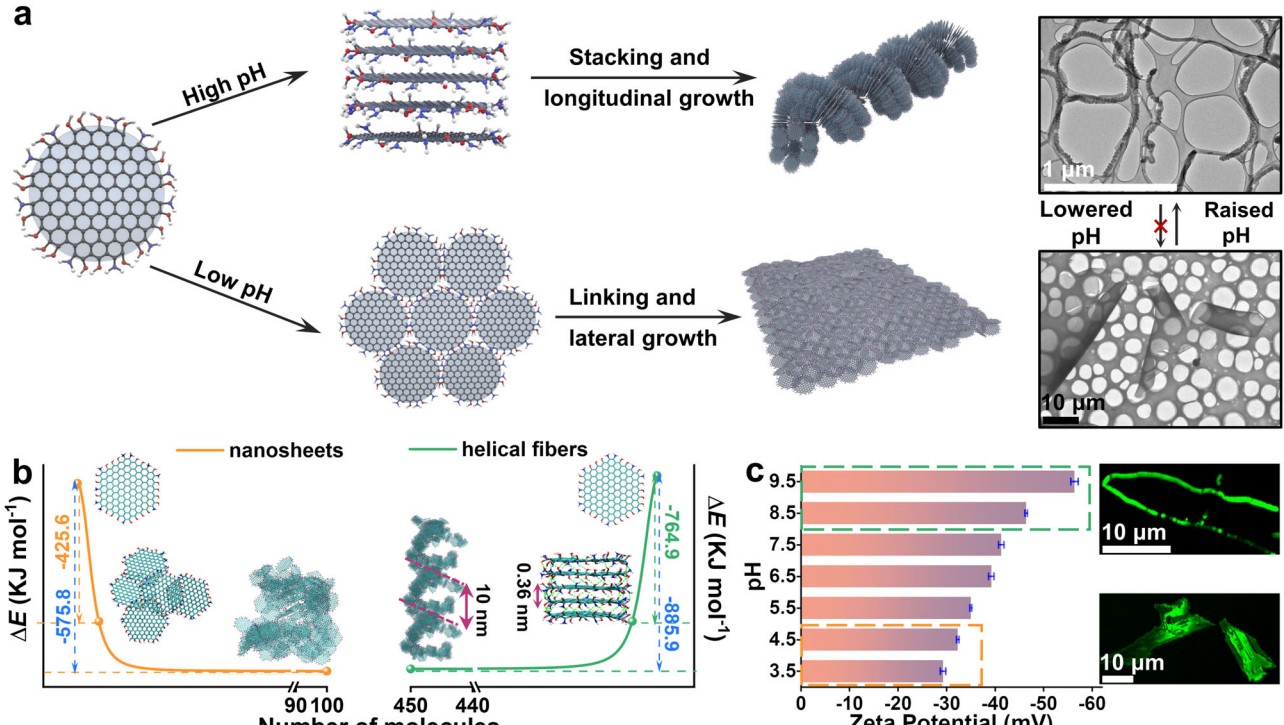

**Fig. 2 | Mechanistic investigation of the GQD-based assemblies. a** Schematic illustration of the NGQD self assembly under different pH. High pH promotes the formation of 1D nuclei, which stack up into helical fibers. Low pH lead to the formation of 2D nuclei, which link with each other into large 2D sheets. Shown in the up right are TEM images of the helical fibers and nanosheets. **b** Molecular

dynamics simulation of the self-assembly process of NGQD at different pH as indicated by the calculated energy profile. **c** ζ potential values of NGQDs-based architectures assembled under various pH conditions. Fluorescent optical micrographs of NGQDs-based architectures at high pH and low pH range.

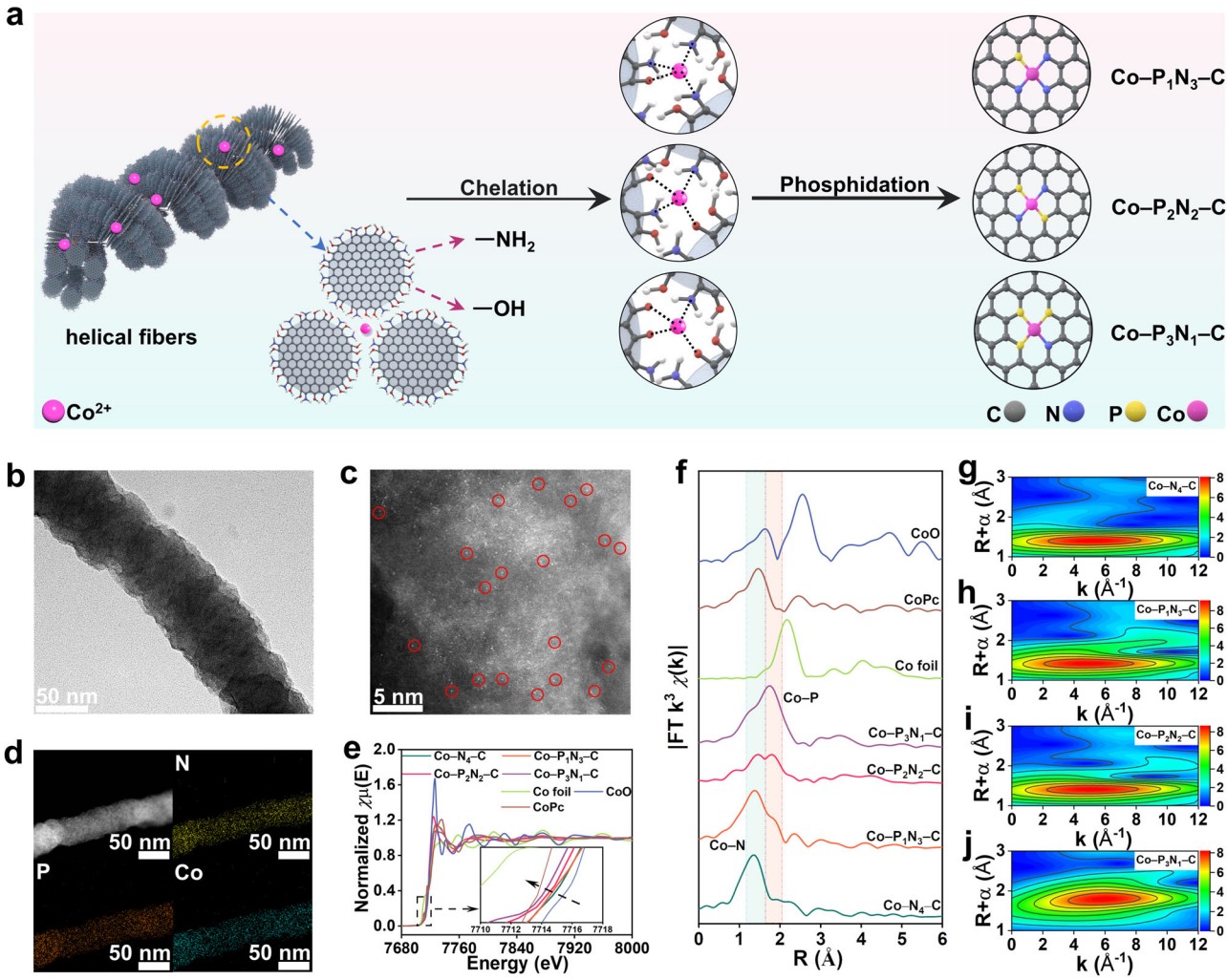

**Fig. 3 | Synthesis and characterization of the Co–P$_x$N$_{4-x}$–C SACs. a** Schematic representation of the preparation of Co–PN–C catalysts. **b** TEM image of single Co–P$_2$N$_2$–C fiber. **c** Atomic-resolution HAADF-STEM image with red circle marked bright dots. **d** HAADF-STEM image and corresponding elemental mapping of Co–P$_2$N$_2$–C. **e** Co K-edge XANES spectra of Co–N$_4$–C, Co–P$_1$N$_3$–C, Co–P$_2$N$_2$–C,

Co–P$_3$N$_1$–C, and reference Co foil, CoO, CoPc. Inset is the enlarged view of the black-dotted region. **f** Fourier-transformed (FT) k$^3$-weighted EXAFS spectra of corresponding catalysts. **g**–**j** EXAFS wavelet transform plots of Co–N$_4$–C, Co–P$_1$N$_3$–C, Co–P$_2$N$_2$–C, and Co–P$_3$N$_1$–C.

to-face π-stacking. This assembly process involved a significant increase in formation energy (ΔE$_5$ = −764.9 kJ/mol) to accommodate more NGQDs into helical-fiber configuration (ΔE$_{450}$ = −885.9 kJ/mol). At pH 3.5, the initial self-assembly of NGQDs exhibited a preference for lateral growth, and resulted in more stable 2D configuration (ΔE$_{100}$ = −575.8 kJ/mol). The formation energy changes during the 100 ns-assembly were shown in Supplementary Fig. 13. Fluorescence confocal imaging of the NGQDs aqueous solution displayed uniform green fluorescence upon excitation at 405 nm at pH 9.5 and pH 3.5 (Supplementary Fig. 14), which transformed to fibers and sheets.

We proposed that the formation of helical fibers or nanosheets at different pH values is influenced by the net charge present on the surface of the NGQDs. At pH values where a high net charge is expected on the assembly, the formation of helical fibers was observed in solution. Conversely, at pH values that partially neutralized the excess surface charge, nanosheets were obtained. Zeta potential (ζ) of the assemblies grown under the test pH conditions has been measured. The ζ value of the assemblies decreases for lower pH of the solution (Fig. 2c). To further trace the transition process, we prepared a series of aqueous solutions with pH gradient from 3.5 to 9.5 to treat the sheet or helical-fiber suspension (Supplementary Fig. 15). As the pH increased, the morphology of the assemblies gradually changed from

2D sheets into helical fibers. However, the transformation from fibers to sheets could not be achieved by adjusting the pH in the reverse direction.

## Integration of helical-fiber into the series of Co–P$_x$N$_{4-x}$–C SACs

The helical fibers were used as a substrate for synthesizing SACs through an assembly-phosphidation approach. Firstly, the chelation between functional groups in the assembly and metal ions (N$_x$···M$^{n+}$···O$_y$) were carried out. The significant decrease in intensity of both N-H and O-H FT-IR signals confirms the chelation between nitrogen, oxygen, and Co$^{2+}$ (Supplementary Fig. 16). Then, the phosphidation transformed this precursor into carbon fiber with P and N co-coordinated single cobalt atoms (Co-PN–C) (Fig. 3a). One can observe that the helical-fiber morphology after phosphidation can be preserved in the temperature range of 500 °C to 700 °C, but collapsed to form fragmented particles at 800 °C (Fig. 3b, Supplementary Fig. 17, denoted as Co-PN–C$_T$). Pyrolysis at either 500 °C or 700 °C yield amorphous oxide/phosphide and Co$_2$(P$_2$O$_7$)/CoP$_2$ crystal particles on carbon (Supplementary Figs. 18 and 19)[29]. For Co-PN–C$_{600}$, no Co-containing nanoparticles were detected in the aberration-corrected high-angle annular dark-field scanning transmission electron microscopy (HAADF-STEM) image (Fig. 3c). The elemental mapping images (Fig. 3d), EDS and EELS spectra

(Supplementary Fig. 20, Fig. 21) reveal the uniform distribution of Co, P, N, and C elements on the fiber. Analysis using inductively coupled plasma mass spectroscopy (ICP-MS) showed that Co–PN–C$_{600}$ (4.41 wt %) and Co–N$_4$–C (4.65 wt%) had similar cobalt content (Supplementary Table 2). Such a topological heteroatom-transfer concept realized by the top-down replacing of the chelated O atom around Co in the precursor by P atom through phosphidation, allows for the transfer of heteroatoms from the assembly into SACs. The most important feature of the topological heteroatom-transfer approach is that the precursor and product share similar helical-fiber structure which is comprised of GQD framework, as well as similar composition except the natures of heteroatoms.

The surface chemical composition of Co–PN–C$_{600}$ was analyzed using XPS (Supplementary Fig. 22). The XPS survey spectra confirm the existence of C, N, P and Co. The deconvoluted C 1$s$ spectra shows $sp^2$ C–C, C–P, C–N peak at 284.8 eV, 285.4 eV and 286.5 eV, respectively (Supplementary Fig. 22b)[30]. The N 1$s$ spectra indicate the coexistence of pyridinic N (397.6 eV), Co–N (398.7 eV), pyrrolic N (400.1 eV), graphitic N (401.1 eV), and oxidized N (402.4 eV)[30], as shown in Supplementary Fig. 22c. The P 2$p$ spectra displays three peaks at 130.5, 132.5 and 133.5 eV, which can be assigned to P–Co, P–C/P–N bonding, and P–O, respectively[31]. As depicted in Co 2$p$ spectrum, the peaks at 780.8 and 797.2 eV could be associated with Co–N/Co–P bond in the Co 2$p_{1/2}$ and 2$p_{3/2}$[32]. Both peaks are shifted to lower values compared to those in Co-N$_4$–C (Supplementary Fig. 23), indicating the increased electron density of Co in Co–PN–C$_{600}$. To further investigate the coordination environment and electronic structure of Co in the Co–PN–C$_{600}$, Co K-edge X-ray absorption spectroscopy (XAS) was employed. X-ray absorption near edge structure (XANES) spectra revealed that the line position of Co–PN–C$_{600}$ and Co–N$_4$–C fell between those of Co foil and CoO (Fig. 3e), which suggests the valence state is between 0 and +2. Compared to Co–N$_4$–C, Co–PN–C$_{600}$ showed a further reduced oxidation state (Supplementary Fig. 24), which could be associated with the lower electronegativity of P compared with N. Specifically, the extended X-ray absorption fine structure (FT-EXAFS) of Co–PN/C$_{600}$ showed two main peaks at $ca.$ 1.8 Å and 1.4 Å assigned to the Co–P and Co–N scattering paths (Fig. 3f)[9]. The absence of Co–Co scattering at $ca.$ 2.2 Å suggests the atomically dispersed nature of the Co sites. Further fitting curves in $R$ and $K$ space indicate the Co atom was coordinated by an equal number of P and N atoms, indicating the presence of Co–P$_2$N$_2$ motifs (Supplementary Fig. 25). As the relative spatial arrangement of P and N in the Co SACs exerts an influence on the coordination environment of Co, which potentially results in distinct catalytic behaviors. We further performed atomic-resolution EELS to evidence the accurate configuration of Co–P$_2$N$_2$ motif (Supplementary Fig. 26). It was observed that the atomically dispersed Co atom is surrounded by two N (green) and two P (blue) atoms in para position.

Both the fundamental study and practical application of SACs require a comprehensive understanding of the structure-activity relationship, which is often unclear due to the challenges in manipulating atomic coordination. While P-coordinated SACs with continuously engineered coordination numbers have been predicted by DFT models[33], they have never been experimentally observed. The feasibility in our proposed strategy for the coordination number manipulation of P in Co–PN–C were demonstrated by adjusting the ratio of –NH$_2$ and –OH (A/H) in the assembling building blocks. Varying this ratio led to the incorporation of different content of O and N in the assemblies for Co ions chelation, which enables the design of the coordination environment of Co–PN–C SACs owing to the preferential substitution of O by P during phosphidation. TEM and atomic scaled HAADF-STEM images reveal that all Co–PN–C SACs shared a similar morphology and Co mass loading (Supplementary Figs. 27 and 28, Table 2). Spectroscopy analysis revealed the corresponding atomic structure as Co–N$_4$–C, Co–P$_1$N$_3$–C, and Co–P$_3$N$_1$–C as the A/H ratio decreases (Supplementary Figs. 25 and 29–33, Table 3). The wavelet

transform spectra indicate that Co–P$_x$N$_{4-x}$ (x = 0–3) exhibit maximum intensities at approximately 4.7, 5.8, 5.2, and 4.9 Å, respectively, which differed significantly from those of Co foil (6.8 Å) and Co$_2$O$_3$ (6.3 Å) (Fig. 3g–j).

## Electrocatalytic hydrogen evolution reaction (HER) performances of the Co–P$_x$N$_{4-x}$–C (x = 0, 1, 2, 3) SACs

The electrochemical activity for the hydrogen evolution reaction (HER) was evaluated for Co–P$_x$N$_{4-x}$–C (x = 0–3) SACs in 1.0 M KOH. Figure 4a displays the HER polarization curves obtained from linear sweep voltammetry (LSV) measurements. Obviously, the HER performance of these Co SACs follows the order: Co–P$_2$N$_2$–C > Co–P$_3$N$_1$–C > Co–P$_1$N$_3$–C > Co–N$_4$–C. Co–P$_2$N$_2$–C exhibits a remarkable overpotential ($\eta_{100}$) of 131 mV to achieve a current density of 100 mA cm$^{-2}$, which is the top performing non-noble-metal SAC for alkaline HER as summarized in Supplementary Table 4. Remarkably, Co–P$_2$N$_2$–C possesses a smallest Tafel slope of 66.9 mV dec$^{-1}$ (Fig. 4b), demonstrating favorable HER kinetics via Volmer-Heyrovsky pathway. Electrochemical impedance spectroscopy (EIS) further manifested a fast HER kinetic process of Co–P$_2$N$_2$–C (Fig. 4c). Similar electrochemical active surface area (ECSA) of Co–P$_x$N$_{4-x}$–C (x = 0–3) SACs indicates that the incorporation of P atoms did not increase the number of active sites[34] (Supplementary Fig. 34). To assess the intrinsic activity of Co–P$_x$N$_{4-x}$–C catalysts, the turnover frequency (TOF) was investigated (Supplementary Fig. 35). At an overpotential of 200 mV, Co–P$_2$N$_2$–C exhibits a higher TOF value of 1.88 s$^{-1}$ compared to the other three catalysts. All these observations suggest that the introduction of coordinated P in Co SACs significantly enhances the electrochemical HER performance as compared with other Co/Ni SACs (Fig. 4d). Furthermore, Co–P$_2$N$_2$–C exhibits remarkable stability with only approximately 5% decay of the current over 20 h operation (Supplementary Fig. 36). It is found that the helix Co–N$_4$–C exhibits superior HER performance compared with non-helix Co–N$_4$–C (Supplementary Figs. 37–39), suggesting the bending surface of helix structure could introduce compressive strain on the supported Co sites, which was beneficial to improve the electrocatalytic activity owing to the distinctive dynamic evolution of Co sites[35,36]. The generalizability of this synthetic approach was verified by the synthesis of Ni SACs, e.g., Ni–P$_1$N$_3$–C (Supplementary Fig. 40, Table 5), which exhibits comparable HER performance with Co–P$_1$N$_3$–C (Supplementary Fig. 41).

To assess its practical application in energy devices, an anion exchange membrane-based (AEM) water electrolyzer was assembled using Co–P$_2$N$_2$–C as the cathode and our recently developed NiFe-MOF@GQD as the anode in a 1.0 M KOH electrolyte (NiFe-MOF@GQD || Co–P$_2$N$_2$–C), as depicted in Fig. 4e. The details of the cell composition are provided in Methods. A reference electrolyser based on RuO$_2$ and Pt/C was built and tested. At a cell voltage of 1.88 V, the NiFe-MOF@GQD || Co–P$_2$N$_2$–C water electrolyser demonstrates the capability to deliver a current density of 1.0 A cm$^{-2}$, outperforming the RuO$_2$ || Pt/C couple, and recently developed water-splitting cells (Supplementary Fig. 42)[37–40], with a remarkable stability at 1.0 A cm$^{-2}$ for 100 h-long stability test (Fig. 4f). The amount of generated hydrogen of Co–P$_2$N$_2$–C was measured during a 60-min electrolysis at constant current of 100 mA, which was higher compared with that of Pt/C (362 vs. 308 μmol, Supplementary Fig. 43a). Faradaic efficiency was determined to be 95.4%, 96.1% and 96.7% for Co–P$_2$N$_2$–C at 1.5 V, 1.6 V and 1.7 V (Supplementary Fig. 43b). A corresponding video of the Co–P$_2$N$_2$–C based AEM electrolyser reveals that at applied cell voltage of 1.7 V, a large amount of H$_2$ bubbles an be observed at the cathode (Supplementary Movie 1).

The detailed structural and compositional evolutions of the Co–P$_2$N$_2$–C after 100-h electrolysis were investigated. The post-HER Co–P$_2$N$_2$–C maintains its helix-fiber morphology without the observation of aggregated particles (Supplementary Fig. 44a, b) with a negligible loss (<1.4 wt%) of Co (Supplementary Table 6). Survey and

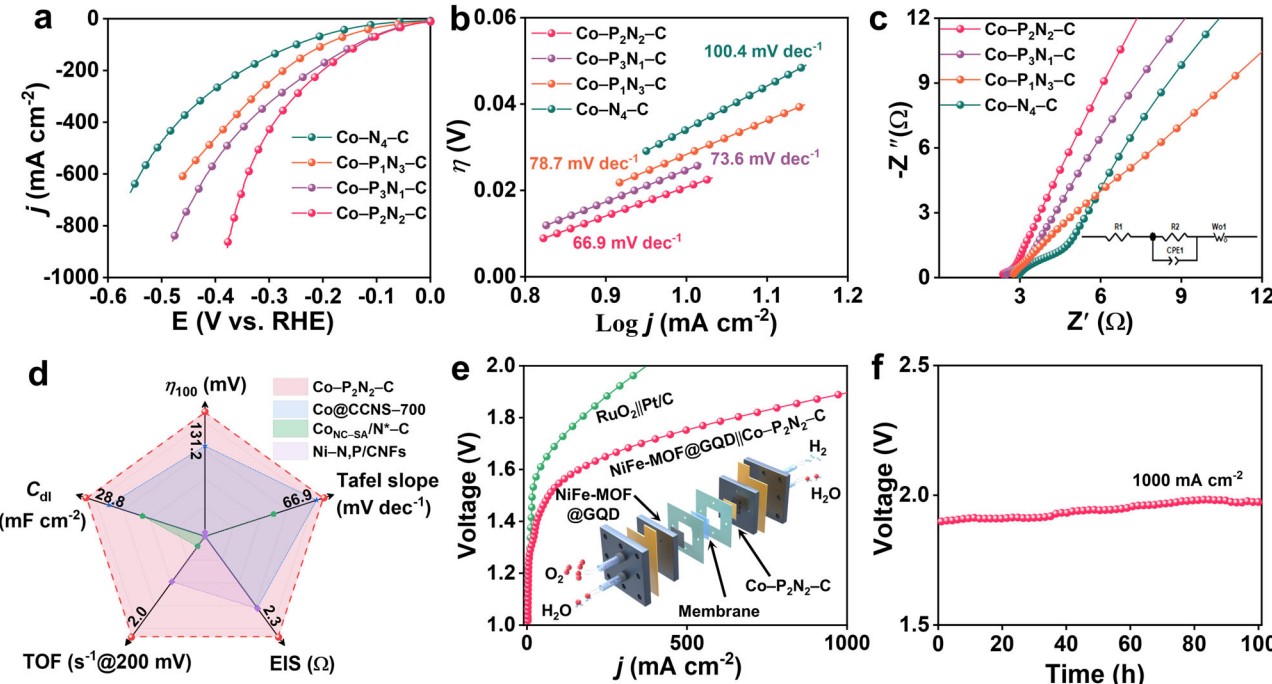

**Fig. 4 | Electrochemical characterization of HER properties in 1.0 M KOH electrolyte (pH = 14). a** Polarization curves. **b** Tafel plots. **c** Nyquist plots of Co−$N_4$−C, Co−$P_1N_3$−C, Co−$P_2N_2$−C, Co−$P_3N_1$−C with mass loading of 0.7 mg cm$^{-2}$ and resistance of 2.9, 2,7, 2.3, and 2.4 Ω, respectively. The scan rate was 5 mV s$^{-1}$. **d** Comparison of the alkaline HER performance with advanced SAC-electrocatalysts in terms of overpotential ($\eta_{100}$), Tafel slope, EIS, TOF, and $C_{dl}$. **e** Polarization curves for overall water splitting of the devices using Co−$P_2N_2$−C and NiFe-MOF@GQD couple, and Pt/C and $RuO_2$ couple, respectively. Inset shows the schematic composition of the two-electrode AEM electrolyzer. **f** Durability test of the Co−$P_2N_2$−C based AEM electrolyser at current density of 1.0 A cm$^{-2}$.

high-resolution XPS spectra maintain the shapes and positions to suggest a stable valence state and chemical environment of the Co single atoms (Supplementary Fig. 44c–g), demonstrating the high structural stability. The long-term stability measurement of HER indicates approximately 2.7%, 3.1%, 3.9%, and 7.4% decay of the current over 100-h operation for Co−$N_4$−C, Co−$P_1N_3$−C, Co−$P_2N_2$−C and Co−$P_3N_1$−C, respectively (Supplementary Fig. 45), following the stability sequence of Co−$N_4$−C > Co−$P_1N_3$−C > Co−$P_2N_2$−C > Co−$P_3N_1$−C.

**Theoretical investigation of the role of phosphorus**

To gain a deeper understanding of how P atoms enhance the HER performance of Co−$P_xN_{4-x}$−C, density functional theory (DFT) calculations were conducted. Computational models for Co−$P_xN_{4-x}$−C (x = 0–3) consist of a graphene layer with a Co atom coordinated by both N and P atoms (Fig. 5a and Supplementary Fig. 46). For Co-$P_2N_2$−C, theoretical calculations were firstly performed to exclude the possibility of formation of two N/P atoms located at adjacent position. The formation energy ($E_{formation}$) of ortho- and para-positioned Co-$P_2N_2$ models indicates that formation of para-positioned Co-$P_2N_2$ is favored owing to a smaller value (−5.83 eV vs. −4.02 eV, Supplementary Table 7). Differential charge density maps (Fig. 5a) demonstrate that the electron-rich state around the Co atom in Co−$P_xN_{4-x}$−C (x = 1, 2, 3) is intensified compared to that of Co−$N_4$−C, indicating lower oxidation states of Co centers and a higher likelihood of contributing electrons to bond with $H_2O$ molecules. Compared with the perfectly symmetrical charge depletion of Co−$N_4$−C, both charge accumulation and depletion occur around Co atoms when there is P coordination, with higher electron cloud density around Co along with the breaking of the symmetry of the charge distribution. Bader charge analysis also confirm the electronic state of Co in Co−$P_xN_{4-x}$−C is 1.01e, 0.85e, 0.67e and 0.47e (Fig. 5a). This lowered charge of Co atom suggests that the negatively charged N draws electrons from Co but the positively charged P donates electron to Co in Co−$P_xN_{4-x}$−C, leading to the decreased valence state of Co with increased coordination number of

P. The overpotential of HER shows a volcanic relationship with the Bader charge of Co atom, which indicates that a moderate coordination number of P in Co−$P_2N_2$−C induces appropriate charge of Co and can lead to optimal HER performance (Supplementary Fig. 47). Notably, $H_2O$ molecules exhibit higher adsorption energies on Co−$P_xN_{4-x}$−C (x = 1, 2, 3) than on Co−$N_4$−C (Supplementary Figs. 46, 48, 49). The projected density of states (PDOS) of Co orbitals in Co−$P_xN_{4-x}$−C (x = 0, 1, 2, 3) gradually shift away from the Fermi level, and the corresponding d-band center undergoes a downshift with the introduction of more coordinated P atoms which suggests a weakened binding strength of intermediates on Co (Supplementary Fig. 50)[41].

According to Tafel slope results, alkaline HER on all Co−$P_xN_{4-x}$−C (x = 0, 1, 2, 3) catalysts follow the Volmer-Heyrovsky process. Stepwise reaction barriers for the alkaline HER were calculated and depicted in Fig. 4b, c. The first sluggish Volmer step is water dissociation ($H_2O$ + e$^-$ = H* + OH$^-$)[42], in which all Co−$P_xN_{4-x}$−C (x = 0, 1, 2, 3) catalysts exhibit uphill formation of enthalpy, with Co−$P_2N_2$−C displaying the smallest Gibbs free energy for water dissociation ($\triangle G_w$) of 1.132 eV. In the second electron transfer step (2H* + 2e$^-$ = $H_2$), the calculations reveal that the dissociated proton preferentially adsorbs on nearby P site rather than N site in Co−$P_2N_2$−C, as indicated by the energy change estimation for $\triangle G_{(H*+OH*)-TS}$ (Supplementary Fig. 51). Additionally, P sites display preferable hydrogen adsorption energy ($\triangle G_{H*}$) compared to N sites in all Co SACs (Fig. 5c), with the most optimized H binding energy ($\triangle G_{H*}$ = 0.069 eV in Fig. 5d and Supplementary Fig. 52) in Co−$P_2N_2$−C. Configurations along the Volmer-Heyrovsky process are presented in Supplementary Fig. 53. According to the (Brønsted−Evans−Polanyi) BEP principles, a low water dissociation barrier requires strong H/OH adsorption on the surface, but which may also result in poisoning of the sites required for water readsorption and hydrogen recombination[43,44]. So the optimal catalyst should balance the energies of water dissociation and H/OH adsorption. The calculated $\triangle G_w$, $\triangle G_{OH*}$ and $\triangle G_{H*}$ demonstrate a volcano relationship with the number of coordinated P atoms in Co SACs, where Co−$P_2N_2$−C possesses optimized values

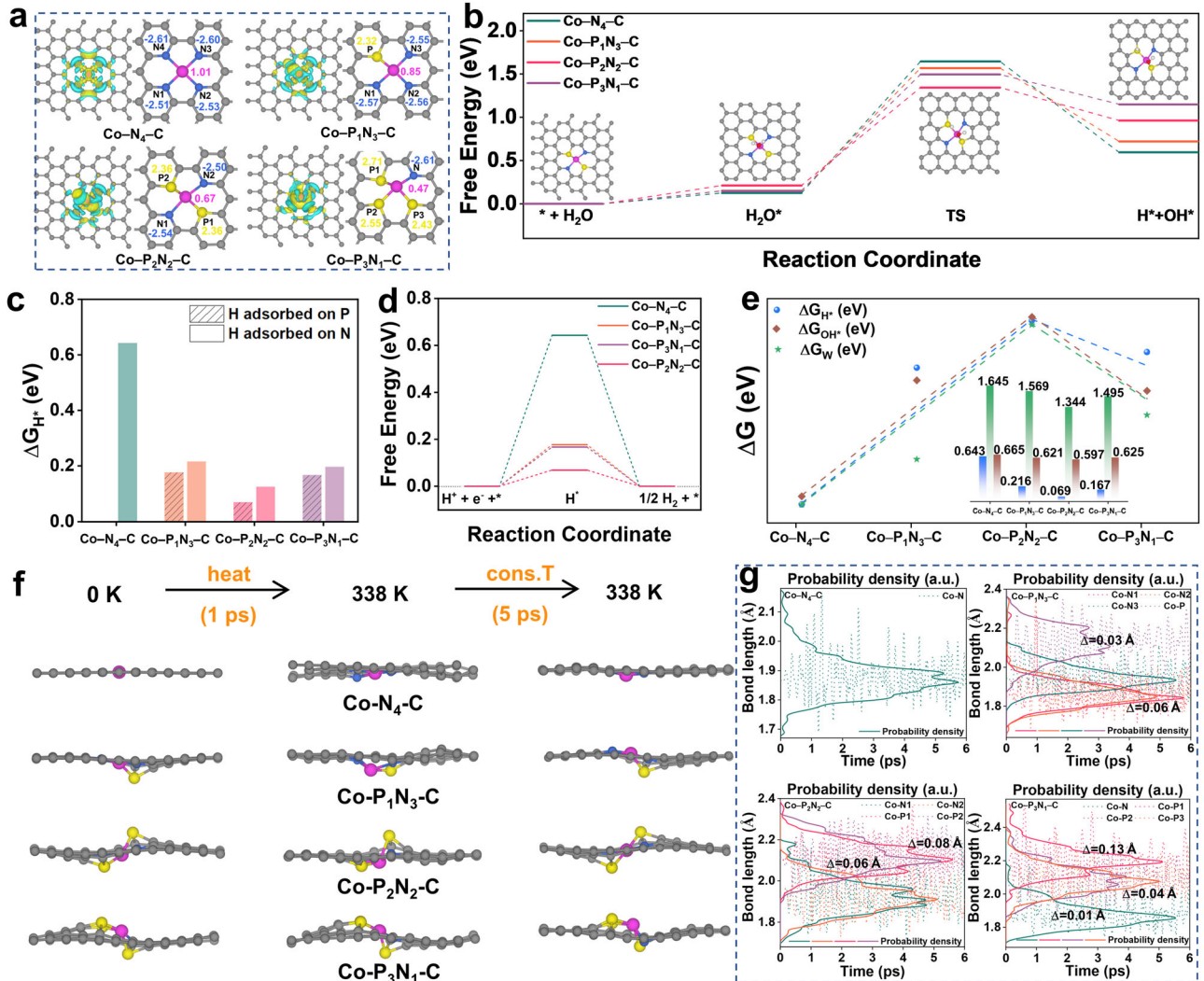

**Fig. 5 | Theoretical investigations. a** Electron density differences top views and Bader charge diagram for Co–N$_4$–C, Co–P$_1$N$_3$–C, Co–P$_2$N$_2$–C, Co–P$_3$N$_1$–C. Blue and yellow represents for electron depletion and accumulation, respectively. **b** HER free energy profile for the water dissociation process. **c** H adsorption energy calculations at the P or N site in Co–P$_x$N$_{4-x}$–C (x = 0, 1, 2, 3). **d** The diagram of $\triangle G_{H^*}$ over corresponding catalysts. **e** The plots of the $\triangle G_{H^*}$, $\triangle G_{OH^*}$ and $\triangle G_w$ as function of different Co–P$_x$N$_{4-x}$–C (x = 0, 1, 2, 3) models. **f** Snapshots of the atomic structure of the Co–P$_x$N$_{4-x}$–C catalysts during the MD simulations at varied temperature. **g** Co–N and Co–P bond length statistics in the Co–P$_x$N$_{4-x}$–C catalysts during MD simulations.

(Fig. 5e). Increasing the coordination number of P in Co–P$_x$N$_{4-x}$–C (x = 0, 1, 2, 3) catalysts causes gradually severe structural distortion and increased electron density around Co atom. Bader charge of the adsorbed H$_2$O molecule on Co–P$_x$N$_{4-x}$–C shows the increased coordination number of P results in higher net Bader charge of H$_2$O adsorbate to involve more electrons in water dissociation (Supplementary Fig. 54). But the increased electron density around Co hinders H$_2$O accumulation owing to stronger electrostatic repulsion, renders an optimized $\triangle G_w$ on Co–P$_2$N$_2$–C. Structural distortion breaks the coordination symmetry, rendering the active site more polar to favor accumulating more OH$^-$ near Co sites, while increasing the electron density around Co favors adverse OH$^-$ desorption owing to stronger electrostatic repulsion. The transferred H would be more effectively accepted on increased P atoms, but encounters difficulties to ensure sufficient electron transfer for further reduction into H$_2$[45]. Therefore, a moderate coordination number of P can optimize the related energies in water dissociation and H/OH adsorption/desorption steps.

We conducted ab initio MD simulations to examine the structural durability of the Co–P$_x$N$_{4-x}$–C catalysts. The thermal vibrations of Co–P$_x$N$_{4-x}$–C could be visualized in the snapshots and quantitatively measured at 0 K and 338 K (Fig. 5f). For all catalysts, no noticeable surface diffusion could be observed. Figure 5g shows the statistics of Co–N and Co–P bonds length during MD, among which the Co–N in Co–N$_4$ has little change while the Co–P bonds experience serve elongation along with increased coordination number of P (Fig. 5g, Supplementary Table 8). DFT calculations were also performed to investigate the binding energy of Co atoms on the support ($E_{binding}$) and the bulk cohesive energy ($E_{cohesive}$). All models are stable against the Co being leached from the support due to the negative $E_{binding}$ values (Supplementary Table 7). A more negative value of $E_{binding}$ - $E_{cohesive}$ means the Co atoms embedding on the support is more stable after long operation, except the positive value of Co–P$_3$N$_1$–C suggesting it is less stable. These calculations suggest that the structure durability decreases with increasing the coordination number of P in Co–P$_x$N$_{4-x}$–C catalysts and are consistent with our experimental results.

## Discussion

In summary, we have proposed a heteroatom-transfer strategy for the synthesis and systematic tailoring of P-coordinated Co single-atom catalysts (SACs) (Co–P$_x$N$_{4-x}$–C, x = 0, 1, 2, 3) by leveraging the self-assembly of graphene quantum dots. Featured with their built-in

functions such as π-conjugation-encapsulated cavity hanged with adjustable functional groups, the structure of the assembly-based precursor can be preserved and the heteroatoms in the chelation environment can be directionally transferred into SACs during phosphidation. Spectroscopy analysis and theoretical calculations have revealed that the incorporation of specifically coordinated P atoms not only leads to a moderate electron density on Co sites, optimizing *OH adsorption in the Volmer step, but also transforms these P atoms into proton-acceptor centers, enhancing the Tafel kinetics and overall HER activity. This work establishes a versatile platform for engineering functional carbon superstructures, and underscores the advantages of precisely tailoring the coordination environment in SACs. We believe that the synthetic and manipulation strategies presented here provide valuable guidance for the design of efficient electrocatalysts in future SAC research.

## Methods

### Chemicals

Pyrene, cobalt (II) acetylacetonate hydrate ($Co(C_5H_7O_2)_2 \cdot xH_2O$), nickel acetylacetonate hydrate ($Ni(C_5H_7O_2)_2 \cdot xH_2O$), sodium hypophosphite ($NaH_2PO_2$) and nafion 117 solution were purchased from Sigma-Aldrich Pte Ltd (Singapore). Nitric acid ($HNO_3$), urea ($CO(NH_2)_2$), potassium hydroxide (KOH), ethanol and citric acid-sodium citrate buffer were purchased from Sinopharm Chemical Reagent Co., Ltd. Pt/C (20 wt.% loading on Vulcan XC-72) and ruthenium oxide ($RuO_2$) were purchased Macklin Biochemical Technology Co., Ltd (Shanghai). Carbon cloth (W0S1011) was purchased from CeTech Co., Ltd. Ni foam was purchased from Changsha Lyrun Material Co., China. All chemicals and used as received without further purification. The water used throughout all experiments was purified using a Millipore system.

### Synthesis of amino-functionalized GQDs (NGQDs)

Typically, pyrene was refluxed in concentrated $HNO_3$ at 80 °C for 12 h with mass concentration of 12.5 g L$^{-1}$. The resulted mixture was diluted and filtered through a 0.22 μm microporous membrane to obtain 1,3,6-trinitropyrene. Then 1,3,6-trinitropyrene (360 mg), sodium bicarbonate (232 mg) and urea (280 mg), were dispersed in water (60 mL) under ultrasonication at 0 °C for 2 h. After that, the suspension was transferred into a Teflon-lined autoclave (100 mL) and heated at 200 °C for 10 h. The resulted suspension was filtered and dialyzed (retained molecular weight: 3500 Da) to remove sodium salt and unfused small molecules and dried to obtain the NGQDs powder.

### Self-assembly of NGQDs to form varied architectures

Such prepared NGQD powder was dispersed in water with concentration of 0.17 mg mL$^{-1}$ under sonication for 30 min. Then the pH of the NGQDs suspension was adjusted with citric acid-sodium citrate buffer (0.1 M) to 9.5 under stirring, and then left standing for 72 h to obtain 1D helical fibers. For the self-assembly of 2D nanosheets, the pH of original NGQDs suspension was adjusted to 3.5. All the assemblies were collected by centrifugation and dried at 60 °C.

### Synthesis of NGQDs-assemblies-based Co single atom catalysts (Co SACs)

Helical fiber was used as-proof-of-concept to be carbon substrate for Co SACs loading. The $Co(C_5H_7O_2)_2$ stock solution (1 mg mL$^{-1}$) was firstly prepared with the dissolve of $Co(C_5H_7O_2)_2 \cdot xH_2O$ in water. Then 1D helical fiber (10 mg), the stock solution (5 mL) were mixed with water (25 mL) under stirring for 10 min to obtain a homogeneous suspension before transfer to a Teflon-lined autoclave and heated at 80 °C for 3 h. After cooling down, the suspension was freeze-dried. Phosphidation was conducted in the tube furnace under a slow argon flow, with $NaH_2PO_2$ (up-stream) as the phosphorus source. The molar ratio of Co to P being 1:6. Then the temperature was raised to 600 °C at a ramping rate of 10 °C min$^{-1}$ and hold for 3 h to obtain Co−PN−C$_{600}$.

Different phosphidation temperature (500 °C, 700 °C, 800 °C) were employed to produce Co−PN−C$_{500}$, Co−PN−C$_{700}$, and Co−PN−C$_{800}$, respectively. To remove undesired metallic species, the products were leached in $H_2SO_4$ (1 M) for 3 h at 80 °C followed by thoroughly washing. The content of Co SACs in carbon was also optimized with varying the volume of $Co(C_5H_7O_2)_2$ stock solutions. For comparison, Co−N$_4$−C was prepared via the same method except the incorporation of $NaH_2PO_2$ during pyrolysis. The synthesis of non-helix Co−N$_4$−C follows the similar procedure of that for helix Co−N$_4$−C with close loading (4.65 wt% vs. 4.61 wt%). The synthesis of Ni SACs follows the similar procedure of that for Co SACs with $Ni(C_5H_7O_2)_2$ salt (1 mg mL$^{-1}$) as stock solution.

### Characterizations

The X-ray diffraction (XRD) patterns were collected using a Bruker D8 Advance with Cu Kα radiation ($\lambda = 1.5406$ Å). The morphology and structure of the catalysts were characterized by a field-emission scanning electron microscopy (FESEM, Zeiss_Supra55), and transmission electron microscopy (TEM, Tecnai G2F30S-TWIN operated at 300 kV). High-angle annular dark field (HAADF) images of the samples were acquired on a JEOL NEOARM200F microscope equipped with probe correctors at 200 kV. STEM-EELS spectrum imaging data was acquired in STEM mode with a dispersion of 0.18 eV/channel and a pixel acquisition time of 0.1 s, and STEM-EELS signal spectrum data was acquired in STEM mode with a dispersion of 0.35 eV/channel and a pixel acquisition time of 0.1 s using the Quantum Gatan Imaging Filter. The EDS and corresponding HAADF-STEM image, and EELS data were acquired on a Thermo Fisher Scientific Titan Themis Z microscope at 300 kV, equipped with a probe corrector and a Gatan GIF Continuum dual-EELS system. X-ray photoelectron spectroscopy (XPS) analysis was performed on an ESCALAB 250Xi using Mg as the excitation source. X-ray absorption near-edge structure (XANES) were measured at the BL14W1 beamline of Singapore Synchrotron Radiation Facility and RapidXAFS 2M. Fourier transform infrared spectra (FITR) were obtained on a PerkinElmer spectrum GX FTIR system. The Raman spectra were collected on a (Renishaw) Raman spectrometer using a 450 nm laser. Inductively coupled plasma optical emission spectrometer (ICP-OES, 5110 ICP-OES, Varian, US) was employed with dissolving the powder sample in acid and diluting the solution. The fluorescence microscopy images were collected on Zess Axio Imager 2 optical microscope with oil-immersion objective lens (×100). Atomic force microscopy (AFM) analysis was conducted on SPM-9700HT. Zeta potential was measured on ZEN3690. The amounts of hydrogen were detected on an online gas chromatography (GC-2014, Shimadzu) using $N_2$ as carrier gas.

### Electrochemical measurements

Electrochemical tests were conducted using a typical three-electrode cell configuration on Autolab 302 N instruments, with a graphite rod and a $Hg/Hg_2Cl_2$ electrode serving as the counter and reference electrode, respectively. The preparation of the working electrodes involves the dispersion of as-prepared catalysts (5 mg) in mixed solvent (400 μL water, 600 μL ethanol, 20 μL 5 wt% Nafion solution) under ultrasonication for 2 h. Such obtained ink was pipetted onto a piece of pretreated carbon fiber paper (0.5 cm$^2$) and then dried in air, with mass loading of 0.7 mg cm$^{-2}$. $N_2$ saturated KOH solution (1.0 M) was chosen as electrolyte for the HER. The polarization curves were collected at a scan rate of 5 mV s$^{-1}$. The Nyquist plots were performed at frequencies ranging from 100000 Hz to 0.1 Hz with an amplitude voltage of 5 mV. Potentials were referenced to a reversible hydrogen electrode (RHE) following the equation: $E_{RHE} = E_{SCE} + 0.059$ pH + 0.242. All electrochemical data was shown with *iR* correction. The equilibrium potential was the zero point of hydrogen evolution reaction (HER)/HOR by using Pt/C as working electrode rotating at 1600 rpm in $H_2$-saturated electrolyte.

## Overall water splitting section

Fumasep FAB-PK-130 membrane was incubated with 1.0 M KOH for 24 h and then washed thoroughly for use. Co–$P_2N_2$–C and Pt/C were used as the cathode catalysts. Typically, the Co–$P_2N_2$–C ink was prepared in the same way described above. Then the ink was spin-coated on Ni foam and dried at 25 °C. Our recently developed NiFe bimetallic organic framework/graphene quantum dot nanoarrays supported on Ti mesh (NiFe-MOF@GQD) was employed as anode. Followed with graphite blocks with single serpentine channel were used for flow channels at both sides, and a stainless-steel endplate and a current collector with the active electrode area is $2 \times 2$ cm$^2$. Prior to AEM testing, 20 cycles of cyclic voltammetry (CV) were performed in the potential window of 1–2 V at a scan rate of 50 mV s$^{-1}$. Then the AEM electrolyser was conducted at 65 °C using $N_2$ saturated KOH solution (1.0 M) as electrolyte with a flowing rate of 20 mL min$^{-1}$.

## Molecular dynamics (MD) simulations

All molecular dynamics simulations were performed using the GROMACS[46] (version 2020.6) simulation package. The CHARMM 36 force field was used to represent GQD and TIP3P water model was used for water, respectively[47]. The pH considers the protonation state of the $NH_2$ group. 500 GQD molecules were first randomly placed in a cubic box of *ca.* 15 nm and then solvated with more than 65000 water molecules. Under the NPT ensemble, the systems required 20 ns-equilibration before production runs of 100 ns. The Nose-Hoover method and the Parrinello-Rahman method maintains the temperature at 298 K and the pressure at 1 atm, respectively. To calculate the non-bonded interactions, a cutoff scheme of 1.2 nm was used. For the long range electrostatic interactions calculations, the Particle Mesh Ewald method[48] with a fourier spacing of 0.1 nm was applied. LINCS algorithm was employed to constraint all the hydrogen atoms-containing covalent bonds[49].

## Density functional theory (DFT) calculations

DFT calculations were performed using the Vienna ab initio simulation package (VASP)[50]. To describe the exchange-correlation functional, the generated gradient approximation (GGA) method in the form of Perdew-Burke-Ernzerhof (PBE) was employed[51]. The D3 approximation was used to account for the van der Waals (vdW) interactions[52]. Calculations were implemented with the cut-off plane wave kinetic energy of 500 eV. To minimize periodical effects, the vacuum layer of 20 Å was considered. Γ−centered Monkhorst-Pack *k* points of $3 \times 3 \times 1$ was chosen for configuration optimization and a denser $7 \times 7 \times 1$ grid was used for electronic properties. All structures were fully optimized when the convergence criterion of electronic energy and force were set to be $10^{-5}$ eV/atom and 0.01 eV/Å, respectively.

The free energy calculations were performed according to the computational hydrogen electrode (CHE) model[53]. The VASP sol code with an implicit solvation model was used to simulate the solvation effect of the electrolyte solution[54], and the relative permittivity of water was set to 80.76. The reaction free energy (Δ*G*) was defined as:

$$\Delta G = \Delta E + \Delta E_{ZEP} - T\Delta S \qquad (1)$$

where Δ*E* is the DFT calculated energy difference of the reaction process, $\Delta E_{ZEP}$ and TΔ*S* are the changes in zero-point energy and entropy between the reaction processes obtained from the vibration frequency calculation of 298.15 K.

The energy of *G*(OH$^-$) can be simulated based on *G*($H_2O$) and *G*(H$^+$) with the equation:

$$G(OH^-) = G(H_2O) - G(H^+) = G(H_2O) - G(1/2H_2) \qquad (2)$$

More information about the Gibbs free energies calculations were shown in Supplementary Information. The kinetics analysis was conducted using the binding energies of $H_2O$, OH and H on Co and P or N sites, respectively.

To examine the stability of Co SACs, *ab* initio molecular dynamics (AIMD) simulation at 338 K was performed with the VASP software. Maximally localized Wannier functions (MLWFs) methodsare employed to calculate the space structures of d-orbitals of single-atom Co subjected to crystal fields[55–57].

## Data availability

The data that support the findings of this study are available within the article and the Supplementary Information and are available from the corresponding authors on request. Source data are provided with this paper.

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

## Acknowledgements

This research is financially supported by the National Natural Science Foundation of China (Grant No. 22102140 (T.J.)), the Natural Science

Foundation of Jiangsu Province (Grant No. BK20211602 (T.J.)), the Six Talent Peaks Project in Jiangsu Province (2019-XCL-101 (J.T.)), the Specially-Appointed Professor Plan in Jiangsu Province (J.T.), the Qing Lan Project of Yangzhou University (T.J.), and the French government aid managed by the Agence Nationale de la Recherche under the France 2030 programme with the reference ANR 22 PEXD 0008 (PEPR DIA-DEM) (R.G.).

## Author contributions

J. T. and T. J. conceived the idea and supervised the project. J. T. and S. Q. designed the experiments. S. Q. synthesized the materials and carried out electrochemical measurements. F. X. and Y. F. carried out some structure characterization and repeated the electrochemical tests. Y. Y designed and performed theoretical calculations. N. C carried out HAADF-STEM and EELS characterizations. J. T., T. J., H. X. and R. G. wrote the paper, with comments from all authors.

## Competing interests

The authors declare no competing interests.
