## [Peer Review File · Nature Communications]

REVIEWER COMMENTS

Reviewer #1 (Remarks to the Author):

In this manuscript, Qian et al. reported on the design and synthesis of a series of Co single-atom catalysts (Co-P_xN_{4-x}-C, where x can be 0, 1, 2, or 3) with precise coordination tailoring. Such an achievement could be realized by using graphene quantum dots as blocks to self-assemble into helical fibers. The use of this nanosized carbon material as building blocks with desired molecular properties toward coordination tailoring in SACs is interesting and efficient. The authors presented comprehensive characterizations to confirm the structures, and demonstrated that the introduction of P atoms in Co-N₄ improved the electrochemical activities for hydrogen evolution reaction both experimentally and theoretically. Among all the developed SACs, Co-P₂N₂-C catalyst exhibits outstanding performance with an overpotential of 131 mV to achieve 100 mA cm⁻². This study provides a comprehensive understanding of the structure-activity relationship in such SACs. I recommend this manuscript for publication in Nature Communications after carefully addressing the following issues.

1. It is a distinguished and novel helix-structure. But the advantages of such helix-structure has not been clarified. Is there any bending effect in the Co SACs?
2. The author defined the coordination configuration of the series of SACs, e.g. Co-P₂N₂ mainly based on fitting of EXAFS. It would be better if the authors could try to get the EELS mapping on N, P to support the structure.
3. It should be evidenced when predicting the generalizability of this strategy. Please give examples among the proposed metals or non-metal heteroatoms. How about their HER performance?
4. In Figure 4f, there are obvious fluctuations and decay in the exhibited stability test. Thus, the surface reconstruction and agglomeration of catalyst seems to be inevitable, which requires more characterizations and discussions.
5. The electrocatalytic HER efficiency is significant to evaluate the practical application of catalysts. However in this work, the generated hydrogen was not quantified, the faradaic (HER) efficiency was not determined.

6. HAADF-STEM images in Figure S24 exhibited some clusters other than single atoms, which is in contradiction with the definition to be SACs.

7. A video of a working AEM water electrolyzer is more intuitive and easy to accept. The authors are suggested to provide such kind of information.

8. As mentioned "potentials were referenced to a reversible hydrogen electrode (RHE)", the zero point of RHE should be carefully calibrated using the HER/HOR equilibrium potential with Pt/C in H₂-saturated solutions.

9. There are some spelling errors such as misspelled "Theata" in Fig. S10e.

Reviewer #2 (Remarks to the Author):

This work developed a kind of SACs with tunable P coordination with Co site which shows good HER performance. Besides a series of experimental work, theoretical calculations were also conducted to reveal the atomic mechanisms of the assembly and improved HER property. However, the discussion of theoretical results is not quite in-depth and persuasive. More detailed analysis and additional simulation are needed before its possible consideration for the publication in Nature Communications. Some questions and comments are as follow:

1. How the number of P coordination affect the durability of catalysts? It should be investigated by experiments or theoretical calculations.

2. Differential charge density maps are presented in Fig. 5a to illustrate the effect of P coordination on electron state around Co site. However, it is lack of the discussion of the relationship between the number of P coordination and the electron state around Co site which is very important to explain why Co-P3N1-C is the best.

3. It is demonstrated based on PDOS and d-band center that the introduction of more coordinated P atoms results in weakened binding strength of intermediates on Co. How does these results account for the different H₂O adsorption ability as presented in Supplementary Fig. 36?

4. Both P and N sites acting as H* adsorption sites were tested in this work. How about Co sites acting as H* adsorption site? Is it energetically unfavorable? Moreover, Are the HER performance of possible bridge sites are tested such as Co-P, Co-N for water adsorption/dissociation, H* adsorption/desorption?

5. Only the comparison of water dissociation and hydrogen adsorption energies is not enough for the fundamental understanding of atomic catalytic performance. Further analysis is indispensable to clarify the mechanisms about How the variation of P coordination optimizes these energies.

6. The caption of Supplementary Fig. 12 seems wrong. (a) should be pH=9.5; (b) should be pH=3.5.

Reviewer #3 (Remarks to the Author):

In this manuscript, the authors have presented a novel approach involving topological heteroatom transfer to precisely modulate the coordination environment of individual Co atoms. Additionally, the effect of A/H ratio and PH values on the self-assembly of graphene quantum dots has been undertaken, revealing the formation mechanism of helical fibers. The phosphidation process, facilitated by the chelation between functional groups and metal ions, promoted the formation of single Co atoms with P and N co-coordination. Among the as-synthesized catalysts, Co-P2N2-C has displayed remarkable catalytic activity and stability, attributed to its optimal adsorption/desorption as verified by DFT calculations. While the overall work is featured by its detail and comprehensiveness, it is imperative to address certain issues pertaining to data analysis and description before considering publication in Nature Communications. Thus, I would recommend a major revision.

1 The alignment between the proposed topological concept and this work appears somewhat limited. Additional elucidation is warranted to underscore this topological concept.

2 In Line 136 and Line 140, it is stated that the peak at 535nm is identified as band B, with a mention of "redshift." However, Figure 1h reveals an initial blue shift in band B for the 24h sample compared to the 0h sample. A meticulous review of the photoluminescence data is recommended.

3 The accurate measurement of interplanar distances raises questions, particularly given the limited resolution of the HRTEM images presented in Figure 1d, Figure S10, and Figure S17.

4 A notable discrepancy arises from the elemental mapping in Figure 3d, which indicates an abundant presence of Co, despite the description on Line 220 stating a Co loading of less than 5 wt%. Furthermore, the absence of the N peak in Figure S19 contradicts the elemental mapping in Figure 3d. These disparities necessitate resolution.

5 The credibility of analysis for XRD patterns in Figures S17 and S18 is compromised due to the low signal-to-noise ratio. Besides, the first three strongest peaks of Co-PN-C700 do not match with the selected standard PDF card (PDF #34-1378).

6 The identification of Co single atoms based on Figure S24 may be a little over-estimated.

7 It is pertinent to inquire whether the relative spatial arrangement of nitrogen (N) and phosphorus (P) in the Co-N₂P₂-C sample exerts an influence on the coordination environment of cobalt (Co), potentially resulting in distinct catalytic behaviors. Especially, whether it is possible for two N/P atoms located at adjacent positions.

8 Given the emphasis on the structural stability of the designed catalyst, it is advisable to include post-TEM characterization of the catalysts after electrochemical measurements.

Responses to reviewers' comments for the manuscript titled "Tailoring coordination environments of single-atom electrocatalysts by topological heteroatom transfer" (NCOMMS-23-45139).

We sincerely thank all reviewers for their positive feedbacks and valuable comments, which shall significantly help to improve the quality of our manuscript. We have thoroughly reviewed and addressed all the points raised by the reviewers, and the changes are highlighted in red in the text. Our point-by-point response to all comments is presented as below, with the reviewers' comments given in blue and our responses in black. We hope that the reviewers are satisfied with these revisions and the manuscript will be accepted for publication in Nature Communications.

Reviewer #1 (Remarks to the Author):

In this manuscript, Qian et al. reported on the design and synthesis of a series of Co single-atom catalysts (Co-P_xN_{4-x}-C, where x can be 0, 1, 2, or 3) with precise coordination tailoring. Such an achievement could be realized by using graphene quantum dots as blocks to self-assemble into helical fibers. The use of this nanosized carbon material as building blocks with desired molecular properties toward coordination tailoring in SACs is interesting and efficient. The authors presented comprehensive characterizations to confirm the structures, and demonstrated that the introduction of P atoms in Co-N₄ improved the electrochemical activities for hydrogen evolution reaction both experimentally and theoretically. Among all the developed SACs, Co-P₂N₂-C catalyst exhibits outstanding performance with an overpotential of 131 mV to achieve 100 mA cm⁻². This study provides a comprehensive understanding of the structure-activity relationship in such SACs. I recommend this manuscript for publication in Nature Communications after carefully addressing the following issues.

Response: We thank the reviewer for the positive comments on our manuscript. We have made extensive revisions based on these suggestions.

1. It is a distinguished and novel helix-structure. But the advantages of such helix-structure has not been clarified. Is there any bending effect in the Co SACs?

Response: We thank the reviewer for the valuable comment. Non-helix fibers were synthesized by tailoring the A/H value to be 0.66 in NGQDs (**Supplementary Fig. 3, Supplementary Table 1, Supplementary Fig. 9b**), as shown in **Fig. 1, Table 1, and Fig. 2** below. The synthesis of non-helix Co–N₄–C follows the similar procedure of that for helix Co–N₄–C, with close loading (4.65 wt% vs. 4.61 wt%, **Supplementary Figs. 37, 38**), as shown in **Fig. 3, Fig. 4** below. It is found that the helix Co–N₄–C exhibits superior HER performance compared with non-helix Co–N₄–C (**Supplementary Fig. 39** shown as **Fig. 5** below), suggesting the bending surface of helix structure could introduce compressive strain on the supported Co sites, which was beneficial to improve the electrocatalytic activity owing to the distinctive dynamic evolution of Co sites (*Nat. Commun.* **12**, 6335 (2021); *J. Am. Chem. Soc.* **136**, 13629–13640 (2014)).

We added this discussion in the Manuscript and figures in Supplementary Information.

Fig. 1 (Supplementary Fig. 3): XPS analysis of NGQDs with different A/H. (a) Survey spectra, (b) C 1s, (c) N 1s, and (d) O 1s spectra.

Table 1 (Supplementary Table 1): Atomic composition analysis. Summary of the atomic content in NGQDs with different A/H.

Samples	C 1s (at.%)	N 1s (at.%)	O 1s (at.%)	A/H
NGQDs	70.83	10.93	18.24	0.69
NGQDs-1	70.73	10.44	18.81	0.66
NGQDs-2	70.81	10.12	19.07	0.61
NGQDs-3	70.58	9.62	19.8	0.54
NGQDs-4	70.72	7.89	21.39	0.43

Fig. 2 (Supplementary Fig. 9): SEM images of the assemblies of NGQDs with varied A/H. (a) 0.69, (b) 0.66, (c) 0.61, (d) 0.54, and (e) 0.43.

Fig. 3 (Supplementary Fig. 37): XPS analysis of non-helix Co-N₄-C. (a) XPS survey spectrum, (b) C 1s, (c) N 1s, (d) Co 2p spectra of non-helix Co-N₄-C. (e) Distribution of N element with different bonding configurations.

Fig. 4 (Supplementary Fig. 38): Characterizations of the non-helix Co-N₄-C. (a) TEM and (b) the high-resolution HAADF-STEM image with spherical aberration correction for non-helix Co-N₄-C. (c) Co K-edge XANES spectra of non-helix Co-N₄-C, and reference Co foil, CoO, CoPc. (d) Corresponding Fourier-transformed (FT) k^3 -weighted EXAFS spectra. (e) The corresponding k^3 -weighted EXAFS fitting curves at k space for the non-helix Co-N₄-C. (f) FT-EXAFS fitting curves of non-helix Co-N₄-C at k space.

Fig. 5 (Supplementary Fig. 39): HER performance of non-helix and helix Co-N₄-C. Polarization curves of non-helix and helix Co-N₄-C in 1.0 M KOH electrolyte. The scan rate was 5 mV s⁻¹.

2. The author defined the coordination configuration of the series of SACs, e.g. Co-P₂N₂ mainly based on fitting of EXAFS. It would be better if the authors could try to get the EELS mapping on N, P to support the structure.

Response: We thank the reviewer for the valuable suggestion. As the relative spatial arrangement of P and N in the Co SACs exerts an influence on the coordination environment of Co, which potentially results in distinct catalytic behaviors. We further performed atomic-resolution EELS to evidence the accurate configuration of Co-P₂N₂ motif (**Supplementary Fig. 26**), as shown in **Fig. 6** below. It is found that the atomically dispersed Co atom is surrounded by two N (green) and two P (red) atoms in para position. Theoretical calculations were also performed to exclude the possibility of formation of two N/P atoms located at adjacent position. The formation energy ($E_{\text{formation}}$) of ortho- and para-positioned Co-P₂N₂ models indicates that formation of para-positioned Co-P₂N₂ is favored owing to a smaller value (-5.83 eV vs. -4.02 eV, **Supplementary Table 7**), as shown in

Table 2 below. Therefore, both experiment and theoretical results confirm that N and P atoms are in the para-position in Co-P₂N₂-C.

We added this discussion in the Manuscript, and figure and table in Supplementary Information.

Fig. 6 (Supplementary Fig. 26): Atomic-resolution EELS analysis of Co-P₂N₂-C. (a) STEM-HAADF image. (b) A synchronous HAADF image acquired with the EELS mapping within the green rectangle. (c) The overlap mapping. Elemental mappings of (d) Co, (e) N, and (f) P.

Table 2 (Supplementary Table 7): Theoretical calculations for the formation energy, binding energy, and cohesive energy of all Co–P_xN_{4-x}. The comparison of $E_{\text{formation}}$, $E_{\text{binding}} - E_{\text{cohesive}}$ values for all Co–P_xN_{4-x} models.

Samples	$E_{\text{formation}}$ (eV)	$E_{\text{binding}} - E_{\text{cohesive}}$ (eV)	Models
Co-N ₄ -C	-8.01	-2.65	Co-P ₁ N ₃ -C	-6.25	-0.99	Co-P ₂ N ₂ -C (para)	-5.83	-0.47	Co-P ₂ N ₂ -C (ortho)	-4.02	1.34	Co-P ₃ N ₁ -C	-5.23	0.13	
3. It should be evidenced when predicting the generalizability of this strategy. Please give examples among the proposed metals or non-metal heteroatoms. How about their HER performance?

Response: We thank the reviewer for the constructive comment. The generalizability of this synthetic approach was verified with the synthesis of Ni SACs, e.g. Ni–P₁N₃–C (Supplementary Fig. 40, Table 5 shown as Fig. 7 and Table 3 below), which exhibits comparable HER performance with Co–P₁N₃–C (Supplementary Fig. 41), as shown in

Fig. 8 below. The synthesis of Ni SACs follows the similar procedure of that for Co SACs with $\text{Ni}(\text{C}_5\text{H}_7\text{O}_2)_2$ salt (1 mg mL^{-1}) as stock solution. More systematic study on other kinds of metal and non-metal heteroatom based SACs is on-going.

We added this discussion in the Manuscript and figures in Supplementary Information.

Fig. 7 (Supplementary Fig. 40): Characterization of Ni-P₁N₃-C catalyst. (a) TEM, (b) HAADF-STEM image. (c) XAFS curves. (c) Ni K-edge XANES spectra of Ni-P₁N₃-C, and reference Ni foil, NiO, NiPc. (d) Fourier-transformed (FT) k^3 -weighted EXAFS spectra of Ni-P₁N₃-C, and reference Ni foil, NiO, NiPc. (e) The k^3 -weighted EXAFS fitting curves at R space for Ni-P₁N₃-C. (f) FT-EXAFS fitting curves of Ni-P₁N₃-C at k space.

Table 3 (Supplementary Table 5): EXAFS fitting of the Ni-P₁N₃-C catalyst. EXAFS fitting parameters at the Ni K-edge various samples ($S_0^2=0.86$).

sample	path	C. N. ^[a]	R (Å) ^[b]	σ^2 ($\times 10^{-3} \text{ \AA}^2$) ^[c]	ΔE (eV) ^[d]	R factor ^[e]
Ni-P ₁ N ₃ -C	Ni-N	3.3±0.5	2.00±0.02	9.5±6.7	-2.5±5.7	0.02
	Ni-P	0.9±0.2	2.31±0.02	7.5±3.6		

^aC. N.: coordination numbers; ^bR: bond distance; ^c σ^2 : Debye-Waller factors; ^d ΔE_0 : the inner potential correction. ^eR factor: goodness of fit.

Fig. 8 (Supplementary Fig. 41): HER performance of Ni-P₁N₃-C: (a) Polarization curve, (b), Tafel plot, (c), Nyquist plot, and (d) Stability test of Ni-P₁N₃-C in 1.0 M KOH electrolyte. The scan rate was 5 mV s⁻¹.

4. In Figure 4f, there are obvious fluctuations and decay in the exhibited stability test. Thus, the surface reconstruction and agglomeration of catalyst seems to be inevitable, which

requires more characterizations and discussions.

Response: We thank the reviewer for raising the critical comment. The detailed structural and compositional evolutions of the Co–P₂N₂–C catalyst after 100-h electrolysis were investigated. The post-HER Co–P₂N₂–C maintains its helix-fiber morphology without the observation of aggregated particles (**Supplementary Fig. 44a, b**) with a negligible loss (< 1.4 wt%) of Co (**Supplementary Table 6**), as shown in **Fig. 9** and **Table 4** below. Survey and high-resolution XPS spectra maintain the shapes and positions to suggest a stable valence state and chemical environment of the Co single atoms (**Supplementary Fig. 44c-g**), demonstrating the high structural stability. The electrolyser maintains 94.9% of initial activity, which could be ascribed to the turbulence caused by bubbles generation and bursting.

We added this discussion in the Manuscript and figures, table in Supplementary Information.

Fig. 9 (Supplementary Fig. 44): Structural characterizations and XPS analysis of the post-HER Co-P₂N₂-C catalyst. (a) TEM, (b) HAADF-STEM image (c) XPS survey spectrum, (d) C 1s, (e) N 1s, (f) P 2p, (g) Co 2p spectra.

Table 4 (Supplementary Table 6): Composition analysis of the post-HER Co–P₂N₂–C catalyst. ICP test of the post-HER Co–P₂N₂–C.

Sample	Co (wt%)
Co–P ₂ N ₂ –C	4.43
Co–P ₂ N ₂ –C (post-HER)	4.37

5. The electrocatalytic HER efficiency is significant to evaluate the practical application of catalysts. However in this work, the generated hydrogen was not quantified, the faradaic (HER) efficiency was not determined.

Response: We thank the reviewer for the nice suggestion. The amount of generated hydrogen of Co–P₂N₂–C was measured during a 60-min electrolysis at constant current of 100 mA, which was higher compared with that of Pt/C (362 vs. 308 μ mol), as shown in **Supplementary Fig. 43a (Fig. 10)** below). Faradaic efficiency was determined to be 95.4%, 96.1% and 96.7% for Co–P₂N₂–C at 1.5 V, 1.6 V and 1.7 V (**Supplementary Fig. 43b**). These results suggest an efficient HER performance of the Co–P₂N₂–C catalyst. The amounts of hydrogen were detected on an online gas chromatography (GC-2014, Shimadzu) using N₂ as carrier gas.

We added this discussion in the Manuscript and figure in Supplementary Information.

Fig. 10 (Supplementary Fig. 43): The electrocatalytic HER efficiency of Co-P₂N₂-C. (a) Hydrogen production at specific current of 100 mA for Co-P₂N₂-C and Pt/C. (b) Faradaic efficiency at a specific voltage (1.5, 1.6, and 1.7 V) of Co-P₂N₂-C and Pt/C. The error bar reflects three measurements.

6. HAADF-STEM images in Figure S24 exhibited some clusters other than single atoms,

which is in contradiction with the definition to be SACs.

Response: We thank the reviewer for pointing out this. Corresponding TEM and the high-resolution HAADF-STEM image with spherical aberration correction for the three materials were updated, with higher magnification to identify the bright contrast spots to be Co single atoms (revised **Supplementary Fig. 27**), as shown in **Fig. 11** below.

Fig. 11 (Supplementary Fig. 27): Atomic-scale characterizations of the Co-PN-C. TEM and the high-resolution HAADF-STEM image with spherical aberration correction for (a-c) Co-N₄-C, (d-f) Co-P₁N₃-C, and (g-i) Co-P₃N₁-C.

7. A video of a working AEM water electrolyzer is more intuitive and easy to accept. The authors are suggested to provide such kind of information.

Response: We thank the reviewer for the nice suggestion. A corresponding movie of the

Co-P₂N₂-C based AEM electrolyser reveals that at applied cell voltage of 1.7 V, a large amount of H₂ bubbles evolve at the cathode (**Supplementary Movie**).

8. As mentioned “potentials were referenced to a reversible hydrogen electrode (RHE)”, the zero point of RHE should be carefully calibrated using the HER/HOR equilibrium potential with Pt/C in H₂-saturated solutions.

Response: We thank the reviewer for the nice suggestion. The equilibrium potential was the zero point of hydrogen evolution reaction (HER)/HOR by using Pt/C as working electrode rotating at 1600 rpm in H₂-saturated electrolyte as shown in **Fig. 12** below.

Fig. 12: Calibration curves for the zero point of RHE. Polarization curves of Pt/C as working electrode rotating at 1600 rpm in H₂-saturated 1.0 M KOH.

9. There are some spelling errors such as misspelled "Theata" in Fig. S10e.

Response: We thank the reviewer for the careful reading. Corresponding spelling errors have been corrected.

Reviewer #2 (Remarks to the Author):

This work developed a kind of SACs with tunable P coordination with Co site which shows good HER performance. Besides a series of experimental work, theoretical calculations were also conducted to reveal the atomic mechanisms of the assembly and improved HER property. However, the discussion of theoretical results is not quite in-depth and persuasive. More detailed analysis and additional simulation are needed before its possible consideration for the publication in Nature Communications. Some questions and comments are as follow:

Response: We are grateful to the reviewer for the thorough review of our manuscript. To fully address the reviewer's concerns on the theoretical section, we have conducted additional experiments, calculations and simulations with more detailed analysis to support our results, which are shown below.

1. How the number of P coordination affect the durability of catalysts? It should be investigated by experiments or theoretical calculations.

Response: We thank the reviewer for the insightful comment. The long-term stability measurement of HER indicates approximately 2.7%, 3.1%, 3.9%, and 7.4% decay of the current over 100-h operation for Co-N₄-C, Co-P₁N₃-C, Co-P₂N₂-C and Co-P₃N₁-C, respectively (**Supplementary Fig. 45** as shown in **Fig. 13** below), following the stability sequence of Co-N₄-C > Co-P₁N₃-C > Co-P₂N₂-C > Co-P₃N₁-C.

We conducted *ab initio* MD simulations to examine the structural durability of the Co-P_xN_{4-x}-C catalysts. The thermal vibrations of Co-P_xN_{4-x}-C could be visualized in the snapshots and quantitatively measured at 0 and 338 K (**Fig. 5f**), as shown in **Fig. 14** below. For all catalysts, no noticeable surface diffusion could be observed. The statistics of Co-N and Co-P bond length during MD show that Co-N in Co-N₄ has little change while Co-P experiences serve elongation along with increased coordination number of P (**Fig. 5g**, **Supplementary Table 8**), as shown in **Fig. 14 and Table 5** below. DFT calculations were also performed to investigate the binding energy of Co atoms on the support (E_{binding}) and the bulk cohesive energy (E_{cohesive}). All models are stable against the Co being leached from the support due to the negative E_{binding} values (**Supplementary Table 7**), as shown in **Table**

6 below. A more negative value of $E_{\text{binding}} - E_{\text{cohesive}}$ means the Co atoms embedding on the support is more stable after long operation, expect the positive value of Co-P₃N₁-C suggests its less stability. These calculations suggest that the structure durability decreases with increasing the coordination number of P in Co-P_xN_{4-x}-C catalysts and is consistent with our experimental results.

We added Fig. 14, a discussion in the revised anuscript, and other figures and tables in Supplementary Information.

Fig. 13 (Supplementary Fig. 45): Durability test of the Co-P_xN_{4-x}-C catalysts. Chronopotentiometry curves of the Co-N₄-C, Co-P₁N₃-C, Co-P₂N₂-C, and Co-P₃N₁-C based AEM electrolyser at current density of 500 mA cm⁻².

Fig. 14 (Fig. 5f, 5g): (f) Snapshots of the atomic structure of the Co-P_xN_{4-x}-C catalysts during the MD simulations at varied temperature. **(g)** Co-N and Co-P bond length statistics in the Co-P_xN_{4-x}-C catalysts during MD simulations.

Table 5 (Supplementary Table 8): The bond length changes during MD simulations.

The static, peak and difference of bond length for Co–N and Co–P in Co–P_xN_{4-x}–C catalysts during MD simulations.

Samples	N/P position	Static bond length (Å)	Peak bond length (Å)	Difference (Å)
Co-N ₄ -C	Co-N	1.8675	1.8675	0
	Co-N1	1.8861	1.9425	0.06
Co-P ₁ N ₃ -C	Co-N2	1.8634	1.8525	-0.01
	Co-N3	1.9358	1.8525	-0.08
	Co-P	2.0742	2.1075	0.03
	Co-N1	1.8944	1.889	-0.005
Co-P ₂ N ₂ -C	Co-N2	1.9136	1.9174	-0.003
	Co-P1	2.0292	2.1132	0.08
	Co-P2	2.1418	2.1994	0.06
	Co-N	1.8529	1.8615	0.01
Co-P ₃ N ₁ -C	Co-P1	2.0407	2.0825	0.04
	Co-P2	2.0692	2.2015	0.13
	Co-P3	2.1906	2.1165	-0.07

Table 6 (Supplementary Table 7): Theoretical calculations for the formation energy, binding energy, and cohesive energy of all Co-P_xN_{4-x}. The comparison of $E_{\text{formation}}$, $E_{\text{binding}} - E_{\text{cohesive}}$ values for all Co-P_xN_{4-x} models.

Samples	$E_{\text{formation}}$ (eV)	$E_{\text{binding}} - E_{\text{cohesive}}$ (eV)	Models
Co-N ₄ -C	-8.01	-2.65	Co-P ₁ N ₃ -C	-6.25	-0.99	Co-P ₂ N ₂ -C (para)	-5.83	-0.47	Co-P ₂ N ₂ -C (ortho)	-4.02	1.34	Co-P ₃ N ₁ -C	-5.23	0.13	
2. Differential charge density maps are presented in Fig. 5a to illustrate the effect of P coordination on electron state around Co site. However, it is lack of the discussion of the relationship between the number of P coordination and the electron state around Co site which is very important to explain why Co-P₃N₁-C is the best.

Response: We thank the reviewer for the valuable comment. Compared with the perfectly symmetrical charge depletion of Co-N₄-C, both charge accumulation and depletion occur around Co atoms when there is P coordination, with higher electron cloud density around Co along with the breaking of the symmetry of the charge distribution. Bader charge

analysis also confirm the electron state of Co in $\text{Co-P}_x\text{N}_{4-x}\text{-C}$ is 1.01e, 0.85e and 0.67e and 0.47e (**Fig. 5a**), as shown in **Fig. 15** below. This lowered charge of Co atom suggests that the negatively charged N draws electrons from Co but the positively charged P donates electron to Co in $\text{Co-P}_x\text{N}_{4-x}\text{-C}$, leading to the decreased valence state of Co with increased coordination number of P. The overpotential of HER shows a volcanic relationship with the Bader charge of Co atom, which indicates that a moderate coordination number of P in $\text{Co-P}_2\text{N}_2\text{-C}$ induces appropriate charge of Co can display the excellent HER performance (**Supplementary Fig. 47**), as shown in **Fig. 16** below.

We added Fig. 15 and discussion in the Manuscript, and Fig. 16 in Supplementary Information.

Fig. 15 (Fig. 5a): Electron density differences top views and Bader charge diagram for $\text{Co-N}_4\text{-C}$, $\text{Co-P}_1\text{N}_3\text{-C}$, $\text{Co-P}_2\text{N}_2\text{-C}$, $\text{Co-P}_3\text{N}_1\text{-C}$. Blue and yellow represents for electron depletion and accumulation, respectively.

Fig. 16 (Supplementary Fig. 47): The volcanoes of HER overpotential for the Bader charge of active center Co atom.

3. It is demonstrated based on PDOS and *d*-band center that the introduction of more coordinated P atoms results in weakened binding strength of intermediates on Co. How does these results account for the different H₂O adsorption ability as presented in Supplementary Fig. 36?

Response: We thank the reviewer for pointing out the critical comment. According to the *d*-band center theory, the shift of *d*-band center is relative to the adjustment of the adsorption properties of intermediates on the active sites. The adsorption energy of H₂O is calculated according to the equation: $E_{\text{ads}} = E_{\text{total}} - E_{\text{catalyst}} - E_{\text{H}_2\text{O}}$, where E_{total} represents the total energy, E_{catalyst} and $E_{\text{H}_2\text{O}}$ are the energy of the catalyst model and the energy of adsorbed H₂O, respectively (revised **Supplementary Fig. 46**, new **Supplementary Fig. 48**), as shown **Fig. 17** and **Fig. 18** below.

It has been previously demonstrated that in TM-N-C system, the *d*-band split into five partial *d* orbitals to selectively participate in the catalytic reactions. Therefore, the energy position of *d*-band center can be orbital dependent and is not sensitive to some structural

perturbation (*J. Phys. Chem. Lett.* **13**, 5969–5976 (2022)). In the Co-P_xN_{4-x}-C catalysts, the local environment of the Co-N₄ undergoes structural deformations when the introduction of P and the adsorbed H₂O is bonded (**Supplementary Fig. 46**, **Supplementary Fig. 48**). Such structural deformation might influence the participation of *d* orbitals, especially frontier orbitals which are closest to the Fermi level of SAC, leading to poor scaling relationship between the adsorption energy of H₂O and *d*-band center (**Fig. 19** below). In detail, the electron density differences show that *d*_{z²} and *d*_{yz} orbitals participate in the formation of Co-O bond in Co-N₄-C and Co-P₂N₂-C, respectively (**Fig. 20** below), which could explain the different H₂O adsorption ability of Co-P_xN_{4-x}-C catalysts (*Nat. Commun.* **13**, 2075 (2022)).

We added Figs. 17 and 18 in Supplementary Information.

Fig. 17 (revised **Supplementary Fig. 46**): Theoretical model of (a) Co-N₄-C, (b) Co-P₁N₃-C, (c) Co-P₂N₂-C, and (d) Co-P₃N₁-C with corresponding energies. The gray, red, blue and yellow balls represent for C, Co, N, and P atoms, respectively.

Fig. 18 (Supplementary Fig. 48): Theoretical models of adsorbed H₂O* on Co-P_xN_{4-x}-C catalysts. The intermediate structures of H₂O* on the model of Co-P_xN_{4-x}-C catalysts with corresponding total energies. The gray, red, blue and yellow balls represent for C, Co, N, and P atoms, respectively.

Fig. 19: DOS for Co-3d orbitals of $\text{Co-P}_x\text{N}_{4-x}\text{-C}$. (a) $\text{Co-N}_4\text{-C}$, (b) $\text{Co-P}_1\text{N}_3\text{-C}$, (c) $\text{Co-P}_2\text{N}_2\text{-C}$, and (d) $\text{Co-P}_3\text{N}_1\text{-C}$.

Fig. 20: Wannier functions of frontier d orbitals of single-atom Co on Co-N₄-C and Co-P₂N₂-C, where yellow (+) and light blue (-) represent the sign of the Wannier function. The electron density differences of active d orbital bonded with H₂O molecules on Co-N₄-C and Co-P₂N₂-C.

4. Both P and N sites acting as H* adsorption sites were tested in this work. How about Co sites acting as H* adsorption site? Is it energetically unfavorable? Moreover, Are the HER performance of possible bridge sites are tested such as Co-P, Co-N for water adsorption/dissociation, H* adsorption/desorption?

Response: We thank the reviewer for the valuable suggestions. The inferior ΔG_{H^*} of Co sites in Co-P₂N₂-C suggests the energetically unfavorable adsorption of H* compared with that of P sites (0.069 eV vs. 0.330 eV, **Supplementary Fig. 52**), as shown in **Fig. 21** below. The HER performance of Co-P and Co-N bridge sites in Co-P₂N₂-C were also evaluated, both H₂O and H* cannot be effectively adsorbed and transfer to the Co sites and P sites for stable adsorption, respectively (**Fig. 22** below), excluding the possibility of these bridge

sites participating in HER.

We added Fig. 21 and corresponding discussion in Supplementary Information.

Fig. 21 (Supplementary Fig. 52): H adsorption energy calculation. The ΔG_{H^*} at the Co site compared with P and N sites in Co-P₂N₂-C.

Fig. 22: Theoretical calculation of HER performance on possible Co-P and Co-N bridge sites in Co-P₂N₂-C. a, H₂O adsorption, and b, H adsorption on Co-P and Co-N bridge sites in Co-P₂N₂-C.

5. Only the comparison of water dissociation and hydrogen adsorption energies is not enough for the fundamental understanding of atomic catalytic performance. Further analysis is indispensable to clarify the mechanisms about How the variation of P

coordination optimizes these energies.

Response: We thank the reviewer for the valuable comment. We revised the discussion “Furthermore, both ΔG_w and ΔG_{H^*} demonstrate a volcano relationship with the number of coordinated P atoms in Co SACs, where Co–P₂N₂–C exhibits the highest HER activity (Fig. 5e). The high performances of Co–P₂N₂–C can be ascribed to the moderate coordination number of P which enables a proper charge distribution on Co–P region with optimal adsorption/desorption of the intermediate. In this material, P atoms serve as new proton-acceptor instead of N.” as follows:

According to the (Brønsted–Evans–Polanyi) BEP principles, a low water dissociation barrier requires strong H/OH adsorption on the surface, but which may also result in poisoning of the sites required for water readsorption and hydrogen recombination (*Nat. Mater.* **11**, 550–557 (2012); *Angew. Chem. Int. Ed.* **57**, 7568–7579 (2018)). So the optimal catalyst should balance the energies of water dissociation and H/OH adsorption. The calculated ΔG_w , ΔG_{OH^*} and ΔG_{H^*} demonstrate a volcano relationship with the number of coordinated P atoms in Co SACs, where Co–P₂N₂–C possesses optimized values (**Fig. 5e**), as shown in **Fig. 23** below. Increasing the coordination number of P in Co–P_xN_{4-x}–C (x = 0, 1, 2, 3) catalysts causes gradually severe structural distortion and increased electron density around Co atom. Bader charge of the adsorbed H₂O molecule on Co–P_xN_{4-x}–C shows the increased coordination number of P results in higher net Bader charge of H₂O adsorbate to involve more electrons in water dissociation (**Supplementary Fig. 54**), as shown in **Fig. 24** below. But the increased electron density around Co hinders H₂O accumulation owing to stronger electrostatic repulsion, renders an optimized ΔG_w on Co–P₂N₂–C. Structural distortion breaks the coordination symmetry, rendering the active site more polar to favor accumulating more OH⁻ near Co sites, while increasing the electron density around Co favors adverse OH⁻ desorption owing to stronger electrostatic repulsion. The transferred H would be more effectively accepted on increased P atoms, but encounters difficulties to ensure sufficient electron transfer for further reduction into H₂ (*Nat. Commun.* **12**, 3783 (2021)). Therefore, a moderate coordination number of P can optimize the related energies in water dissociation and H/OH adsorption/desorption steps.

We added Fig. 23 and discussions in Manuscript, and Fig. 24 in Supplementary Information.

Fig. 23 (Fig. 5e): The plots of the ΔG_{H^*} , ΔG_{OH^*} and ΔG_W as function of different Co-P_xN_{4-x}-C (x=0, 1, 2, 3) models.

Fig. 24 (Supplementary Fig. 54): Bader charge analysis. The calculated Bader charge of adsorbed H₂O over Co-P_xN_{4-x}-C.

6. The caption of Supplementary Fig. 12 seems wrong. (a) should be pH=9.5; (b) should be pH=3.5.

Response: We thank the reviewer for the careful reading. The caption of **Supplementary Fig. 12** has been revised as: “**Supplementary Fig. 13: Formation energy and snapshots of self-assembly during 100 ns simulation period. (a) pH=9.5 for helical fibers and (b) pH=3.5 for nanosheets.**”

Reviewer #3 (Remarks to the Author):

In this manuscript, the authors have presented a novel approach involving topological heteroatom transfer to precisely modulate the coordination environment of individual Co atoms. Additionally, the effect of A/H ratio and PH values on the self-assembly of graphene quantum dots has been undertaken, revealing the formation mechanism of helical fibers. The phosphidation process, facilitated by the chelation between functional groups and metal ions, promoted the formation of single Co atoms with P and N co-coordination. Among the as-synthesized catalysts, Co-P₂N₂-C has displayed remarkable catalytic activity and stability, attributed to its optimal adsorption/desorption as verified by DFT calculations. While the overall work is featured by its detail and comprehensiveness, it is imperative to address certain issues pertaining to data analysis and description before considering publication in Nature Communications. Thus, I would recommend a major revision.

Response: We are grateful to the reviewer for the constructive comments and suggestions about our manuscript. To fully address the reviewer’s concerns, we have conducted additional experiments and characterizations, especially pertaining to data analysis and description to substantially revise our manuscript, which are shown below.

1. The alignment between the proposed topological concept and this work appears somewhat limited. Additional elucidation is warranted to underscore this topological concept.

Response: We thank the reviewer for the valuable suggestion. Such a topological

heteroatom-transfer concept realized by the top-down replacement of the chelated O atom around Co in the precursor by P atom through phosphidation, allows for the transfer of heteroatoms from the assembly into SACs. The most important feature of the topological heteroatom-transfer approach is that the precursor and product share similar helical-fiber structure built of GQD framework, as well as similar composition except the kinds of heteroatoms.

We added this discussion in the Manuscript.

2. In Line 136 and Line 140, it is stated that the peak at 535 nm is identified as band B, with a mention of "redshift." However, Figure 1h reveals an initial blue shift in band B for the 24 h sample compared to the 0 h sample. A meticulous review of the photoluminescence data is recommended.

Response: We thank the reviewer for the pointing out this issue. We repeated the self-assembly of NGQDs at pH = 9.5 and collected time-dependent PL data (**Supplementary Fig. 7** shown as **Fig. 25** below), which showed similar result as that in Fig. 1h. As we described the observation of the red shift in band B during self-assembly, we mainly focused on the comparison of the final and initial state, that is the 72 h sample and 0 h sample, but ignored the intermediates (24 h and 48 h sample). Actually, there is a 4-7 nm blue shift in band B for 24 h sample (Fig. 1h, and **Supplementary Fig. 7a**), which is relative to the increased twisted deformation induced photoluminescence change (see corresponding SEM images in **Supplementary Fig. 7b**). Such a twisted deformation is believed to be due to a slight change in the molecular packing with a slight separation of central sp^2 domains upon molecular sliding from the most stable *J*-aggregated state (*CrystEngComm.* **23**, 5763-5767 (2021); *Angew. Chem. Int. Ed.* **58**, 5614-5618 (2019)).

We added the figure and discussion in Supplementary Information.

Fig. 25 (Supplementary Fig. 7): Time-dependent PL spectra and SEM images during the self-assembly of NGQDs. (a) PL spectra, (b) SEM images during the self-assembly of NGQDs into helix-fibers at pH=9.5.

3. The accurate measurement of interplanar distances raises questions, particularly given the limited resolution of the HRTEM images presented in Figure 1d, Figure S10, and Figure S17.

Response: We thank the reviewer for the valuable comment. High-resolution HRTEM images were shown in Fig. 1d, Supplementary Fig. 11c and Fig. 18 (Fig. 26, Fig. 27, Fig. 28), respectively, to ensure the accurate measurement of interplanar distances accordingly.

Fig. 26 (Fig. 1d): high-resolution TEM image of the helical-fibers.

Fig. 27 (Supplementary Fig. 11c): HRTEM image of the nanosheets.

Fig. 28 (Supplementary Fig. 18a, b): (a) TEM and (b) HRTEM image of Co-PN-C₅₀₀.

4. A notable discrepancy arises from the elemental mapping in Figure 3d, which indicates an abundant presence of Co, despite the description on Line 220 stating a Co loading of less than 5 wt%. Furthermore, the absence of the N peak in Figure S19 contradicts the elemental mapping in Figure 3d. These disparities necessitate resolution.

Response: We thank the reviewer for insightful comment and pointing out this. We present

the characterization details in the Characterizations section as follows: High-angle annular dark field (HAADF) images of the samples were acquired on a JEOL NEOARM200F microscope equipped with probe correctors at 200 kV. STEM-EELS spectrum imaging data was acquired in STEM mode with a dispersion of 0.18 eV/channel and a pixel acquisition time of 0.1s, and STEM-EELS signal spectrum data was acquired in STEM mode with a dispersion of 0.35 eV/channel and a pixel acquisition time of 0.1s using the Quantum Gatan Imaging Filter on a Thermo Fisher Scientific Titan Themis Z microscope at 300 kV, equipped with a probe corrector and a Gatan GIF Continuum dual-EELS system.

The elemental mapping of Co–P₂N₂–C in Fig. 3d reflects the total signals of the Co–P₂N₂–C loaded on ultrathin carbon coated TEM grid. It is found that corresponding EDS analysis of the sample consists intensive peaks of C and Cu due to the background signals contributed by the substrate (**Supplementary Fig. 20** shown as **Fig. 29** below), which might enhance the overall signal in elemental mapping. The elemental mapping after background subtraction is shown (**Fig. 3d** shown as **Fig. 30** below), which reveals a uniform but relative weak signals of Co, P, and N elements distributed on the fiber. An enlargement of the peak of N in the EELS spectrum was added (**Supplementary Fig. 21** shown as **Fig. 31** below). To examine the presence of N signal more accurately, a low-resolution EELS spectrum to give a whole view of the elemental composition on the fiber is collected (**Fig. 32**), which reflects a much enhanced N signal.

We revised Fig. 3d in the Manuscript and added Figs. 29 and 30 in Supplementary Information.

Fig. 29 (Supplementary Fig. 20): EDS spectra for elemental mapping in Fig. 3d.

Fig. 30 (Fig. 3b, c, d): **b**, TEM image of single Co-P₂N₂-C fiber. **c**, Atomic-resolution HAADF-STEM image. **d**, HAADF-STEM image and elemental mapping of Co-P₂N₂-C.

Fig. 31 (Supplementary Fig. 21): Elemental characterization. The whole EELS spectrum of Co-PN-C₆₀₀ with an enlargement of N peak.

Fig. 32: Elemental characterization. Elemental characterization. A low-resolution EELS spectrum of Co-PN-C₆₀₀.

5. The credibility of analysis for XRD patterns in Figures S17 and S18 is compromised due to the low signal-to-noise ratio. Besides, the first three strongest peaks of Co-PN-C700 do not match with the selected standard PDF card (PDF #34-1378).

Response: We thank the reviewer for the careful reading and pointing out this. We performed XRD characterizations for both samples on the same equipment with a much slower scan rate and revised the analysis and discussion. Pyrolysis at 500 °C yield amorphous oxide/phosphide on carbon, with characteristic diffraction peaks at 29.4° and 41.6° correspond to the graphitic carbon (**Supplementary Fig. 18c**), as shown in **Fig. 33** below. Pyrolysis at 700 °C yield $\text{Co}_2(\text{P}_2\text{O}_7)/\text{CoP}_2$ crystal particles on carbon, with characteristic diffraction peaks at 15.2°, 23.3°, 29.5°, 30.2°, 34.3°, and 17.6°, 24.6°, 35.6° correspond to the crystal plane of $\text{Co}_2(\text{P}_2\text{O}_7)$ (JCPDS No. 84-2126), and CoP_2 (JCPDS No.77-0263), respectively (**Supplementary Fig. 19c**), as shown in **Fig. 34** below.

We revised the figures and added the discussion in Supplementary Information.

Fig. 33 (Supplementary Fig. 18c): XRD pattern of Co-PN-C₅₀₀.

Fig. 34 (Supplementary Fig. 19c): XRD pattern of Co-PN-C₇₀₀.

6. The identification of Co single atoms based on Figure S24 may be a little over-estimated.

Response: We thank the reviewer for the nice suggestion. Corresponding TEM and the high-resolution HAADF-STEM image with spherical aberration correction for the three materials were updated, with higher magnification to identify the bright contrast spots to be Co single atoms (**Supplementary Fig. 27**), as shown in **Fig. 35** below.

Fig. 35 (Supplementary Fig. 27): Atomic-scale characterizations of the Co-PN-C. TEM and the high-resolution HAADF-STEM image with spherical aberration correction for **(a-c)** Co-N₄-C, **(d-f)** Co-P₁N₃-C, and **(g-i)** Co-P₃N₁-C.

7. It is pertinent to inquire whether the relative spatial arrangement of nitrogen (N) and phosphorus (P) in the Co-N₂P₂-C sample exerts an influence on the coordination environment of cobalt (Co), potentially resulting in distinct catalytic behaviors. Especially, whether it is possible for two N/P atoms located at adjacent positions.

Response: We thank the reviewer for the valuable comment. As the relative spatial arrangement of P and N in the Co SACs exerts an influence on the coordination environment of Co, which potentially results in distinct catalytic behaviors. We further performed atomic-resolution EELS to evidence the accurate configuration of Co-P₂N₂ motif (**Supplementary Fig. 26**), as shown in **Fig. 36** below. It is found that the atomically dispersed Co atom is surrounded by two N (green) and two P (blue) atoms in para position.

Theoretical calculations were also performed to exclude the possibility of formation of two N/P atoms located at adjacent position. The formation energy ($E_{\text{formation}}$) of ortho- and para-positioned Co-P₂N₂ models indicates that formation of para-positioned Co-P₂N₂ is favored owing to a smaller value (-5.83 eV vs. -4.02 eV, **Supplementary Table 7**), as shown in **Table 7** below. Therefore, both experiment and theoretical results confirm that N and P atoms are in the para-position in Co-P₂N₂/C.

We added the figure and table in Supplementary Information and corresponding discussions in Manuscript.

Fig. 36 (Supplementary Fig. 26): Atomic-resolution EELS analysis of Co-P₂N₂-C. (a) STEM-HAADF image. (b) A synchronous HAADF image acquired with the EELS mapping within the green rectangle. (c) The overlap mapping. Elemental mappings of (d) Co, (e) N, and (f) P.

Table 7 (Supplementary Table 7): Theoretical calculations for the formation energy, binding energy, and cohesive energy of all Co–P_xN_{4-x}. The comparison of $E_{\text{formation}}$, $E_{\text{binding}} - E_{\text{cohesive}}$ values for all Co–P_xN_{4-x} models.

Samples	$E_{\text{formation}}$ (eV)	$E_{\text{binding}} - E_{\text{cohesive}}$ (eV)	Models
Co-N ₄ -C	-8.01	-2.65	Co-P ₁ N ₃ -C	-6.25	-0.99	Co-P ₂ N ₂ -C (para)	-5.83	-0.47	Co-P ₂ N ₂ -C (ortho)	-4.02	1.34	Co-P ₃ N ₁ -C	-5.23	0.13	
8. Given the emphasis on the structural stability of the designed catalyst, it is advisable to include post-TEM characterization of the catalysts after electrochemical measurements.

Response: We thank the reviewer for the nice suggestion. The detailed structural and compositional evolutions of the Co–P₂N₂–C after 100-h electrolysis were investigated. The post-HER Co–P₂N₂–C maintains its helix-fiber morphology without the observation of aggregated particles (**Supplementary Fig. 44a, b**) with a negligible loss (< 1.4 wt%) of Co (**Supplementary Table 6**), as shown in **Fig. 37** and **Table 8** below. Survey and high-resolution XPS spectra maintain the shapes and positions to suggest a stable valence state and chemical environment of the Co single atoms (**Supplementary Fig. 44c-g**), demonstrating the high structural stability.

We added the figure and table in Supplementary Information and discussions in Manuscript.

Fig. 37 (Supplementary Fig. 44): Structural characterizations and XPS analysis of the post-HER Co-P₂N₂-C catalyst. (a) TEM, (b) HAADF-STEM image (c) XPS survey spectrum, (d) C 1s, (e) N 1s, (f) P 2p, (g) Co 2p spectra.

Table 8 (Supplementary Table 6): Composition analysis of the post-HER Co-P₂N₂-C catalyst. ICP test of the post-HER Co-P₂N₂-C.

Sample	Co (wt%)
Co-P ₂ N ₂ -C	4.43
Co-P ₂ N ₂ -C (post-HER)	4.37

REVIEWER COMMENTS

Reviewer #1 (Remarks to the Author):

This manuscript has been well revised and largely improved, and can be published now.

Reviewer #2 (Remarks to the Author):

The response to the proposed comments is persuasive and the manuscript is well revised. I recommend this revised version to be published on Nature Communications.

Reviewer #3 (Remarks to the Author)

In the response letter and revised manuscript, most of the questions have been well answered with detailed analysis. However, there are still some issues that need to be illustrated further before considering publication in Nature Communications.

1.To enhance the clarity of Fig.1, the revised manuscript should include the elucidation of the blue shift observed in the intermediates ascribed to twisted deformation.

2.The C and Cu signals originate from the Cu grid, which is covered by carbon film. As shown in Supplementary Fig. 20, there are no overlap spectra for Co and Cu, so how could the Cu grid contribute to enhancing the signal of Co? And how do the authors conduct the background subtraction regarding the discrepancy between the revised EDS mapping and the initial version? Additionally, the quantitative component data derived from EDS analysis is expected to be integrated with the EDS spectrum, providing a direct-viewing elemental composition.

3.The intensity of the strongest peak in the XRD pattern of Co-PN-C700(Supplementary Fig. 19c) doesn't align with the Co₂(P₂O₇) phase (PDF#84-2126). Further analysis is expected to verify the phase.

**Responses to reviewers' comments for the manuscript titled "Tailoring coordination environments of single-atom electrocatalysts by topological heteroatom transfer"
(NCOMMS-23-45139A).**

We sincerely appreciate the reviewers for their positive feedback and valuable comments regarding our manuscript. The point-to-point responses to the reviewer's concerns are outlined below, with the reviewer's comments are given in blue and our responses are in black text. Corresponding changes are highlighted by red in the text. We hope these revisions meet the reviewer's expectations and the manuscript will be accepted for publication in Nature Communications.

REVIEWER #1 (Remarks to the Author):

This manuscript has been well revised and largely improved, and can be published now.

Response: We sincerely thank the reviewer for the positive feedback and support for the publication of our study.

REVIEWER #2 (Remarks to the Author):

The response to the proposed comments is persuasive and the manuscript is well revised. I recommend this revised version to be published on Nature Communications.

Response: We sincerely thank the reviewer for the positive feedback and encouraging of the publication of our study.

REVIEWER #3 (Remarks to the Author):

In the response letter and revised manuscript, most of the questions have been well answered with detailed analysis. However, there are still some issues that need to be illustrated further before considering publication in Nature Communications.

Response: We sincerely thank the reviewer for the positive feedback and valuable suggestions regarding our work. To address the remaining issues, we supplemented experiments about STEM elemental mapping with corresponding EELS and EDS spectra, and detailed analysis and discussion of the XRD data, which are shown below.

1. To enhance the clarity of Fig.1, the revised manuscript should include the elucidation of the blue shift observed in the intermediates ascribed to twisted deformation.

Response: We thank the reviewer for the valuable suggestion. The elucidation of the blue shift observed in the intermediates ascribed to the twisted deformation has been added to the discussion of Fig. 1 in the Manuscript accordingly: *“As we described the observation of the red shift in band B during self-assembly, we mainly focused on the comparison of the final and initial state, that is the 72 h sample and 0 h sample. Actually, there is a 4-7 nm blue shift in band B for 24 h sample (Fig. 1h and Supplementary Fig. 7a), which is relative to the increased twisted deformation induced photoluminescence change, corresponding SEM images are shown (Supplementary Fig. 7b). Such a twisted deformation is believed to be due to a slight change in the molecular packing with a slight separation of central sp^2 domains upon molecular sliding from the most stable J-aggregated state”*.

2. The C and Cu signals originate from the Cu grid, which is covered by carbon film. As shown in Supplementary Fig. 20, there are no overlap spectra for Co and Cu, so how could the Cu grid contribute to enhancing the signal of Co? And how do the authors conduct the background subtraction regarding the discrepancy between the revised EDS mapping and the initial version? Additionally, the quantitative component data derived from EDS analysis is expected to be integrated with the EDS spectrum, providing a direct-viewing elemental composition.

Response: We thank the reviewer for the critical comments. We apologize for the misunderstandings caused by our unclear explanation in our previous response. What we intended to convey was that the signal of Co would be stronger than the actual value when the continuous bremsstrahlung X-rays background signal is not subtracted (Continuous bremsstrahlung X-rays are generated when high energy electrons are interacted and slowed by electrostatic interaction with the nucleus of the studied materials). However, we agree with the reviewer that the signal-to-noise in the previous background corrected data is not sufficient to draw a conclusion on the elemental distribution. Based on the reviewer's comment, we acquired new data on a gold grid without carbon coating and performed basic background correction as implemented in the Velox software (Thermo Fisher Scientific) to obtain a more reliable elemental distribution signal and to enhance the signal-to-noise ratio of the data (**Fig. 3d** as shown **Fig. 1** below, and **Fig. 2** below). The EDS and corresponding HAADF-STEM image, and EELS data were acquired on a Thermo Fisher Scientific Titan Themis Z microscope at 300 kV, equipped with a probe corrector and a Gatan GIF Continuum dual-EELS system. The EDS background was fitted by a polynomial function in windows in between the main signal peaks (peaks of the selected elements). The EDS elemental distribution maps were denoised by an average filter with a kernel size of 3 pixels. The acquired EDS and EELS spectra were shown in **Supplementary Figs. 20** and **21 (Fig. 5** and **Fig. 6** below).

To provide a direct-viewing elemental composition, the quantitative component data derived from EDS analysis is shown inset the spectrum (**Supplementary Fig. 20**), which indicates the rough content of C, Co, N, P and O is around 61.87 wt%, 5.68 wt%, 5.42 wt%, 6.51 wt% and 20.51 wt%, respectively. The Au and Cu signals are contributed by the Au grid and the Cu tip of the holder, respectively. The small peaks of K and Ca are impurities induced during the sample preparation steps. The content of Co shows slight discrepancies between the EDS quantification data and the ICP results, which may arise from the uneven distribution of Co in the sample.

We revised these figures and discussions in the Manuscript and Supplementary Information.

Fig. 1 (Fig. 3d): HAADF-STEM image and corresponding elemental mapping of Co-P₂N₂-C.

Fig. 2: Raw data of HAADF-STEM image and corresponding elemental mapping of Co-P₂N₂-C. Raw data before background signal subtraction.

Fig. 3 (Supplementary Fig. 20): EDS spectrum of the corresponding elemental mapping in Fig. 3d (inset: the enlarged spectrum within the energy range of 0-1 keV and the quantification table).

Fig. 4 (Supplementary Fig. 21): Elemental characterization. The EELS spectrum of Co-PN-C₆₀₀.

3. The intensity of the strongest peak in the XRD pattern of Co-PN-C700 (Supplementary Fig. 19c) doesn't align with the Co₂(P₂O₇) phase (PDF#84-2126). Further analysis is expected to verify the phase.

Response: We appreciate the reviewer's careful evaluation of the XRD data. It is important to note that, for the identification of a phase by powder XRD, the position of the XRD peaks is the primary criteria. The intensities of XRD peaks can be strongly influenced by many parameters which cannot be controlled (except for bulk powder samples) such as the preferential orientation or/and the morphology of the phase (i.e. larger or smaller in some directions). In our case, the characteristic diffraction peaks at 15.2°, 23.3°, 29.5°, 30.2°, 34.3° are well assigned to (011), (111), (122), (102) and (013) planes of $\text{Co}_2(\text{P}_2\text{O}_7)$ (JCPDS No. 84-2126), and the ones located at 17.6°, 24.6°, 35.6° are assigned to (100), (111) and (200) planes of CoP_2 (JCPDS No.77-0263), respectively (**Supplementary Fig. 19c**), as shown in **Fig. 5** below.

In addition, the strong peak at 23.3° might also have for origin the carbon substrate with interplanar spacing of 0.xx nm corresponding to the (120) plane (JCPDS No. 50-0926) (*J. Electroanal. Chem.* **904**, 115882 (2022)), as shown in **Supplementary Fig. 19b** (**Fig. 6** below).

We revised these figures and supplemented the discussion in Supplementary Information.

Fig. 5 (Supplementary Fig. 19c): XRD pattern of Co-PN-C₇₀₀.

Fig. 6 (Supplementary Fig. 19b): HRTEM image of Co-PN-C₇₀₀.

REVIEWERS' COMMENTS

Reviewer #3 (Remarks to the Author):

In the response letter, all the questions have been well answered. I recommend this manuscript be published in Nature Communications after minor revision.

1. Please ensure the correct image size of the EDS mapping results in Fig.3d.
2. Regarding the explanation of XRD fitting, the authors attribute the strongest intensity of (111) to preferential orientation or/and the morphology of the phase, which is deemed reasonable. To further support this assumption, it is suggested to provide TEM images to demonstrate the preferential (111) orientation of Co₂(P₂O₇). Additionally, if feasible, the XRD pattern of the carbon substrate obtained through the same treatment should be included to substantiate the authors' hypothesis regarding the peak at 23.3°.

Responses to reviewer's comments for the manuscript titled "Tailoring coordination environments of single-atom electrocatalysts by topological heteroatom transfer" (NCOMMS-23-45139B).

We sincerely appreciate the reviewer for the positive feedback regarding our manuscript. The point-to-point responses to the reviewer's comments are outlined below, with the reviewer's comments given in blue and our responses are in black text. Corresponding changes are highlighted by red in the main text. We hope these revisions meet the your expectations and the manuscript will be accepted for publication in Nature Communications.

Reviewer #3 (Remarks to the Author):

In the response letter, all the questions have been well answered. I recommend this manuscript be published in Nature Communications after minor revision.

Response: We sincerely thank the reviewer for the positive feedback and support for the publication of our study. We supplemented XRD experiment, and revised the image size of Fig. 3d, which are shown below.

1. Please ensure the correct image size of the EDS mapping results in Fig.3d.

Response: We thank the reviewer for the careful reading. The image size of the EDS mapping results in Fig. 3d has been revised, as shown in **Fig. 1** below.

Fig. 1 (Fig. 3d): HAADF-STEM image and corresponding elemental mapping of Co-P₂N₂-C.

2. Regarding the explanation of XRD fitting, the authors attribute the strongest intensity of (111) to preferential orientation or/and the morphology of the phase, which is deemed reasonable. To further support this assumption, it is suggested to provide TEM images to demonstrate the preferential (111) orientation of Co₂(P₂O₇). Additionally, if feasible, the XRD pattern of the carbon substrate obtained through the same treatment should be included to substantiate the authors' hypothesis regarding the peak at 23.3°.

Response: We thank the reviewer for the good suggestion. We have made efforts to obtain high-resolution TEM (HRTEM) images of Co₂(P₂O₇). However, it has been observed that the sample is prone to damage when subjected to beam illumination at high magnification under our testing conditions. Considering that HRTEM images of this sample is not deemed indispensable, we have opted not to pursue further attempts. The XRD pattern of carbon substrate synthesized under the same treatment of that for Co-PN-C₇₀₀ was collected (**Fig. 2** below), which suggests a diffraction peak located at 23.4°, confirming the possible origin of the peak from carbon substrate in Supplementary Fig. 19c.

We revised the Fig. 3d in the Manuscript.

Fig. 2: XRD pattern of carbon substrate synthesized at 700 °C.